# Coding and regulatory variants are associated with serum protein levels and disease

Valur Emilsson [1,2,11 ✉], Valborg Gudmundsdottir [1,11], Alexander Gudjonsson[1,11], Thorarinn Jonmundsson [2], Brynjolfur G. Jonsson[1], Mohd A. Karim[3,4], Marjan Ilkov[1], James R. Staley[5], Elias F. Gudmundsson [1], Lenore J. Launer [6], Jan H. Lindeman[7], Nicholas M. Morton [8], Thor Aspelund [1], John R. Lamb[9], Lori L. Jennings [10] & Vilmundur Gudnason [1,2 ✉]

Circulating proteins can be used to diagnose and predict disease-related outcomes. A deep serum proteome survey recently revealed close associations between serum protein networks and common disease. In the current study, 54,469 low-frequency and common exome-array variants were compared to 4782 protein measurements in the serum of 5343 individuals from the AGES Reykjavik cohort. This analysis identifies a large number of serum proteins with genetic signatures overlapping those of many diseases. More specifically, using a study-wide significance threshold, we find that 2021 independent exome array variants are associated with serum levels of 1942 proteins. These variants reside in genetic loci shared by hundreds of complex disease traits, highlighting serum proteins' emerging role as biomarkers and potential causative agents of a wide range of diseases.

[1] Icelandic Heart Association, Holtasmari 1, IS-201 Kopavogur, Kopavogur, Iceland. [2] Faculty of Medicine, University of Iceland, 101 Reykjavik, Reykjavík, Iceland. [3] Wellcome Trust Sanger Institute, Welcome Genome Campus, Hinxton, Cambridgeshire CB10 1SA, UK. [4] Open Targets, Wellcome Genome Campus, Hinxton, Cambridgeshire CB10 1SD, UK. [5] BHF Cardiovascular Epidemiology Unit, Department of Public Health and Primary Care, University of Cambridge, Cambridge, UK. [6] Laboratory of Epidemiology and Population Sciences, Intramural Research Program, National Institute on Aging, Bethesda, MD 20892-9205, USA. [7] Department of Surgery, Leiden University Medical Center, Leiden, Netherlands. [8] Centre for Cardiovascular Science, Queen's Medical Research Institute, University of Edinburgh, Edinburgh EH16 4TJ, UK. [9] GNF Novartis, 10675 John Jay Hopkins Drive, San Diego, CA 92121, USA. [10] Novartis Institutes for Biomedical Research, 22 Windsor Street, Cambridge, MA 02139, USA. [11] These authors contributed equally: Valur Emilsson, Valborg Gudmundsdottir, Alexander Gudjonsson. ✉email: valur@hjarta.is; v.gudnason@hjarta.is

Large-scale genome-wide association studies (GWASs) have expanded our knowledge of the genetic basis of complex disease. As of 2018, approximately 5687 GWASs have been published revealing 71,673 DNA variants to phenotype associations[1]. Furthermore, exome-wide genotyping arrays have linked rare and common variants to many complex traits. For example, 444 independent risk variants were recently identified for lipoprotein fractions across 250 genes[2]. Despite the overall success of GWAS, the common lead single nucleotide polymorphisms (SNPs) rarely point directly to a clear causative polymorphism, making determination of the underlying disease mechanism difficult[3–6]. Regulatory variants affecting mRNA and/or protein levels and structural variants like missense mutations can point directly to the causal candidate. Alteration of the amino acid sequence may affect protein activity and/or influence transcription, translation, stability, processing, and secretion of the protein in question[7–9]. Thus, by integrating intermediate traits like mRNA and/or protein levels with genetics and disease traits, the identification of the causal candidates can be enhanced[3–6].

Proteins are arguably the ultimate players in all life processes in disease and health, however, high throughput detection and quantification of proteins has been hampered by the limitations of available proteomic technologies. Recently, a custom-designed Slow-Off rate Modified Aptamer (SOMAmer) protein profiling platform was developed to measure 4782 proteins encoded by 4137 human genes in the serum of 5457 individuals from the Age, Gene/Environment Susceptibility Reykjavik Study (AGES-RS)[10], resulting in 26.1 million individual protein measurements. Various metrics related to the performance of the proteomic platform including aptamer specificity, assay variability, and reproducibility have already been described[10]. We demonstrated that the human serum proteome is under strong genetic control[10], in line with findings of others applying identical or different proteomics technologies[11,12]. Moreover, serum proteins were found to exist in regulatory groups of network modules composed of members synthesized in all tissues of the body, suggesting that system-level

coordination or homeostasis is mediated to a significant degree by thousands of proteins in blood[13]. Importantly, the deep serum and plasma proteome is associated with and prognostic for various diseases as well as human life span[10,14–20].

In this work, we regressed levels of 4782 proteins on 54,469 low-frequency and common variants from the HumanExome BeadChip exome array, in sera from 5343 individuals of the deeply phenotyped AGES-RS cohort. Further cross-referencing of all significant genotype-to-protein associations to hundreds of genetic loci for various disease endpoints and clinical traits, demonstrated profound overlap between the genetics of circulating proteins and disease-related phenotypes. We highlight how triangulation of data from different sources can link genetics, protein levels, and disease(s), with the intention of cross-validating one another and pointing to the potential causal relationship between proteins and complex disease(s).

## Results

Using genotype data from an exome array (HumanExome BeadChip) enriched for structural variants and tagged for many GWAS risk loci (Methods), the effect of low-frequency and common variants on the deep serum proteome was examined. Quality control filters[21] and exclusion of monomorphic variants reduced the available variants to 76,891. Additionally, we excluded variants at minor allele frequency (MAF) < 0.001 as they provide insufficient power for single-point association analysis[22]. This resulted in 54,469 low-frequency (54%, MAF < 0.05) and common variants (46%, MAF ≥ 0.05) that were tested for association to each of the 4782 human serum protein measurements using linear regression analysis adjusted for the confounders age and sex (Methods). The current platform targets the serum proteome arising largely from active or passive secretion, ecto-domain shedding, lysis, and/or cell death[10,23]. Figure 1 highlights the classification of the protein population targeted by the aptamer-based profiling platform, showing over 70% of the proteins are secreted or single-pass transmembrane receptors.

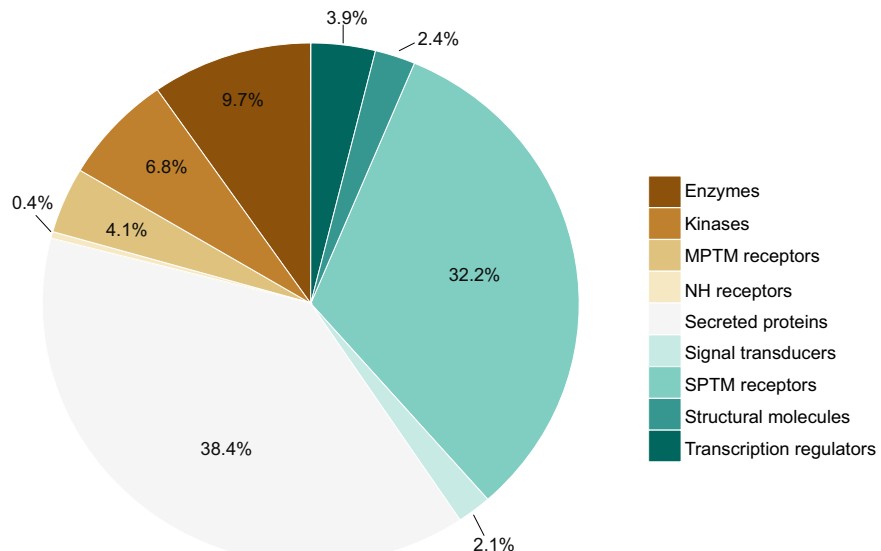

**Fig. 1 Classification of the target protein population.** The pie chart shows the relative distribution (percentage) of the different protein classes targeted by the present proteomics platform (4137 unique proteins), with secreted proteins (38.4%) and single-pass transmembrane (SPTM) receptors (32.2%) dominating the target protein population. Protein classes were manually curated based on information from the SecTrans, Gene Ontology (GO), and Swiss-Prot databases, and were composed of secreted proteins (e.g., cytokines, adipokines, hormones, chemokines, and growth factors), SPTM receptors (e.g., tyrosine and serine/threonine kinase receptors), multi-pass transmembrane (MPTM) receptors (e.g., GPCR, ion channels, transporters), enzymes (intracellular), kinases, nuclear hormone receptors (NH receptors), structural molecules, transcriptional regulators and signal transducers.

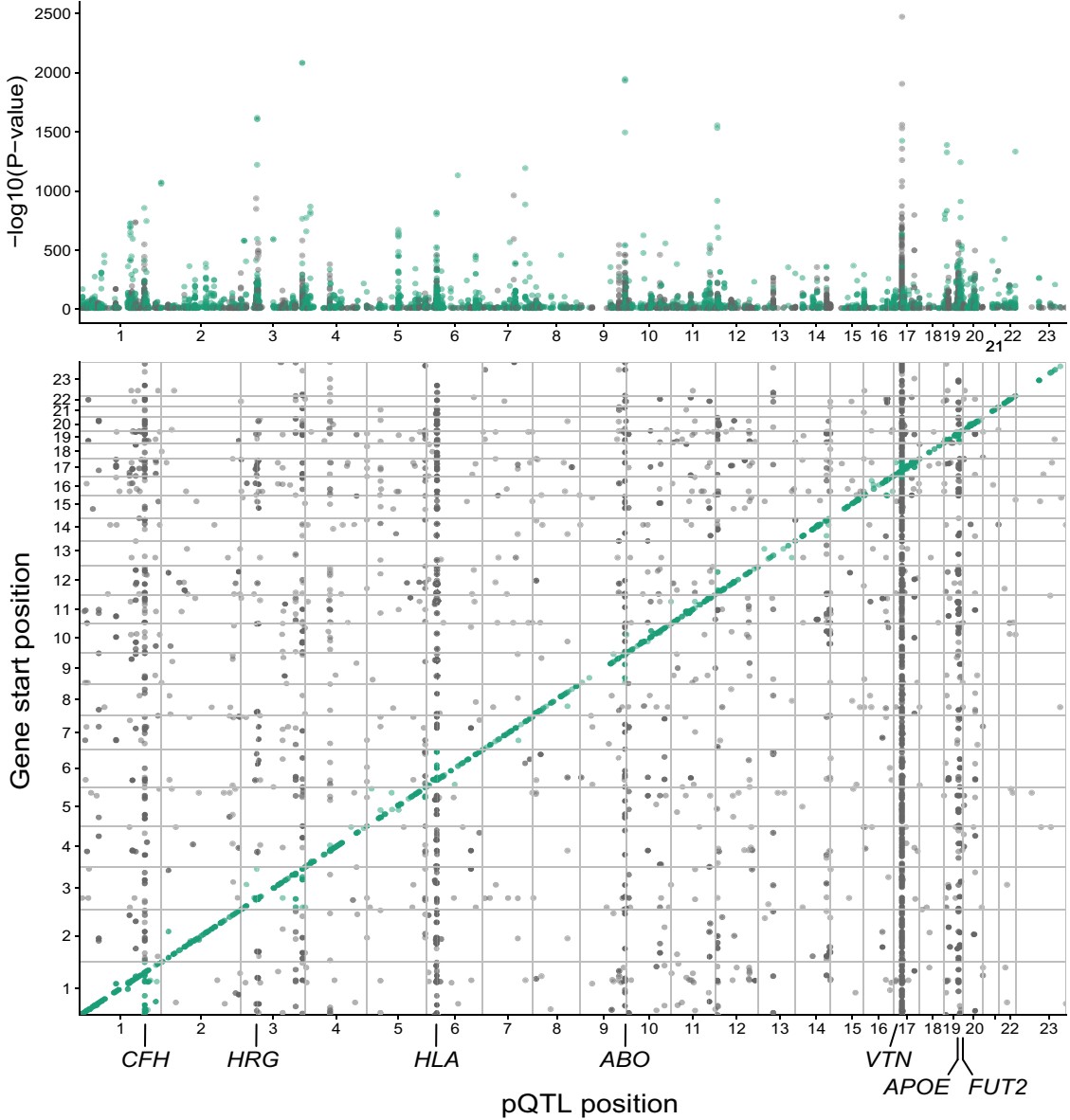

**Fig. 2 A graphical representation of all pQTL discoveries in the current study.** The Manhattan plot in the top panel uses precise two-sided $P$-values as $-\log(P\text{-value})$ for the association (linear regression) of low-frequency and common exome array variants to 4782 proteins in serum. The bottom panel shows the genomic locations of all study-wide significant pQTLs (linear regression, $P < 1.92 \times 10^{-10}$, two-sided), also shown in Supplementary Data 1, where the start position of the protein-encoding gene is shown on the $y$-axis and the location of the pSNP at the $x$-axis. *Cis* acting effects, using a 300 kb window, appear at the diagonal while *trans* acting pQTL effects including *trans* hot spots show up off-diagonally. The genetic loci highlighted across the $x$-axis are *trans*-acting hotspots.

Applying a Bonferroni corrected significance threshold of $P < 1.92 \times 10^{-10}$ (0.05/54469/4782) we detected 5451 exome array variants that were associated with variable levels of 1942 (2138 aptamers) serum proteins (Supplementary Data 1 and Fig. 2), of which 2021 exome variants were independent affecting 1942 (2135 aptamers) proteins (Supplementary Data 2). Supplementary Data 1 lists all associations at $P$-value $< 1 \times 10^{-6}$, or 10,200 exome array variants affecting 2780 (3104 aptamers) human proteins. These protein quantitative trait loci (pQTLs) were *cis* and/or *trans* acting including several *trans* acting hotspots with pleiotropic effects on multiple co-regulated proteins (Fig. 2). Secreted proteins were enriched for pQTLs ($P$-value $< 0.0001$) as compared to non-secreted proteins using 10,000 permutations to obtain the empirical distribution of the $\chi^2$ test of equality of proportions (Supplementary Fig. 1). This implies that secreted proteins are subject to different, and possibly stronger, genetic control than other proteins identified by the current platform. Supplementary Data 3 summarizes various pathogenicity prediction scores for all independent study-wide significant pQTLs in Supplementary Data 2, using the Ensembl Variant Effect Predictor (VEP)[24,25]. Next, we cross-referenced all the 5451 study-wide significant pQTLs with a comprehensive collection of genetic loci associated with diseases and clinical traits from the curated PhenoScanner database[26], revealing that 60% of all pQTLs were linked to at least one disease-related trait (Supplementary Data 4). We have shown in our previous studies that genetic loci affecting several serum proteins exhibit pleiotropy in relation to complex diseases[10]. An example of a possible pleiotropic effect mediated by the variant rs2251219 within the gene *PBRM1* affecting multiple proteins and sharing genetics with

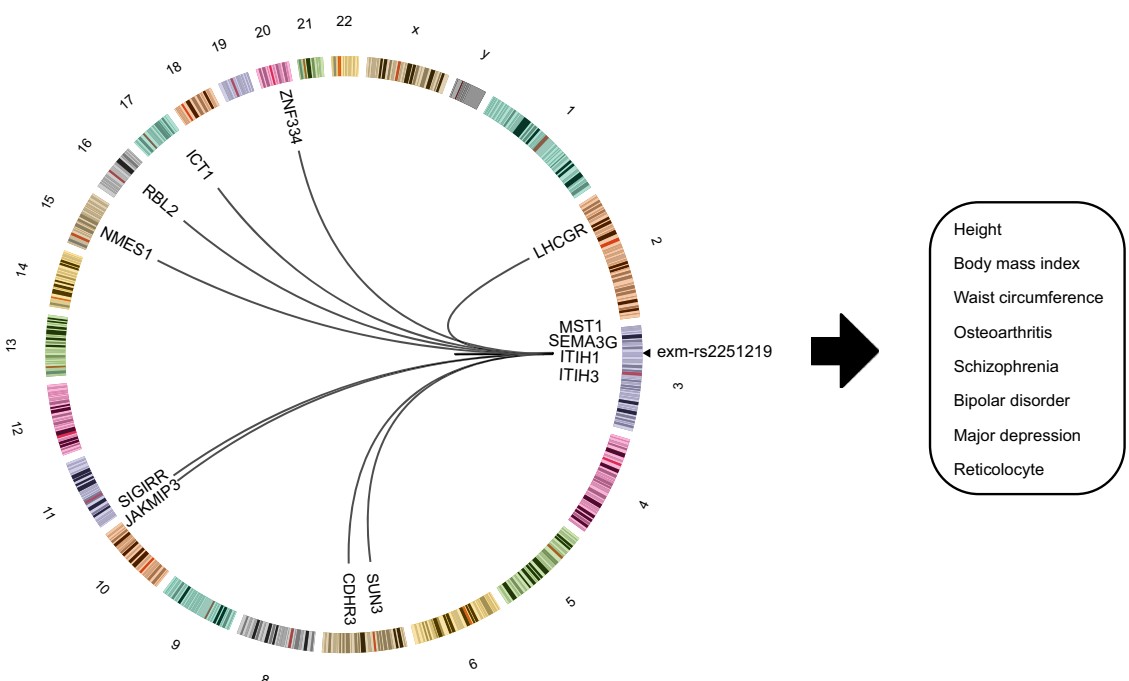

**Fig. 3 Pleiotropy of rs2251219 affects many proteins and disease traits.** The Circos plot highlights the effect of the variant rs2251219 (Supplementary Data 1 and 2) on 13 proteins acting in *cis* or *trans* and sharing genetics with various diseases of different etiologies. Only study-wide significant ($P < 1.92 \times 10^{-10}$, two-sided) genotype-to-protein associations (linear regression) are shown. Lines going from rs2251219 show links to genomic locations of the protein-encoding genes associated with the variant while numbers refer to chromosomes. The arrow points to disease-related traits that have previously been linked to rs2251219.

various diseases and clinical features is illustrated in Fig. 3. Supplementary Figure 2 depicts the relationship between all proteins and some quantitative traits associated with rs2251219. Table 1 highlights a selected set of pQTLs that share genetics with diseases of different etiologies including disorders of the brain, metabolism, immune, cardiovascular system, and cancer. In the sections that follow, we give examples of serum pQTLs that overlap disease risk loci and demonstrate how different data sources can cross-validate one another. Although data triangulation can be used to infer directional consistency, it cannot tell whether the relationship is causal or reactive to a given outcome. As a result, we used two-sample Mendelian randomization (MR) analysis on highlighted examples to test support for a protein's causality to an outcome.

Variable levels of the anti-inflammatory protein TREM2 were associated with two distinct genomic regions (Fig. 4a, Supplementary Fig. 3). This included the missense variant rs75932628 (NP_061838.1: p.R47H) in *TREM2* at chromosome 6 (Fig. 4b), known to confer a strong risk of late-onset Alzheimer's disease (LOAD)[27]. The variant was also associated with IGFBPL1 ($P = 3 \times 10^{-18}$) in serum (Supplementary Data 1), a protein recently implicated in axonal growth[28]. Intriguingly, the region at chromosome 11 associated with soluble TREM2 levels harbors variants adjacent to the genes *MS4A4A* and *MS4A6A* including rs610932 known to influence genetic susceptibility for LOAD[29] (Table 1 and Fig. 4a, b). The variant rs610932 was also associated with the proteins GLTPD2 and A4GALT (Supplementary Data 1). The alleles increasing the risk of LOAD for both the common variant rs610932 and the low-frequency variant rs75932628 were associated with low levels of soluble TREM2 (Fig. 4b). Consistently, we find that the high-risk allele for rs75932628 was associated with accelerated mortality post-incident LOAD in the AGES-RS (Fig. 4c). It is of note

that the levels of TREM2 in the cerebrospinal fluid (CSF) reflect the activity of brain TREM2-triggered microglia[4,30], while high levels of CSF TREM2 have been associated with improved cognitive functioning[31]. Supplementary Figure 4 highlights the correlation (Spearman rank) between the different proteins affected by the LOAD risk loci at chromosomes 6 and 11. The accumulated data show a directionally consistent effect at independent risk loci for LOAD converging on the same causal candidate TREM2. Furthermore, a two-sample MR analysis using genetic instruments across the *TREM2* and *MS4A4A/MS4A6A* loci and GWAS associations for LOAD in Europeans as outcome[32] provided evidence that variable TREM2 protein levels are causally related to LOAD ($P = 5.3 \times 10^{-5}$) (Fig. 4d and Supplementary Data 5). The instrument rs7232 (Fig. 4d), an independent variant associated with TREM2 (Supplementary Data 2), is a missense variant in MS4A6A that has previously been linked to LOAD (Supplemental Data 4), but the MS4A cluster has recently been shown to modulate the production of soluble TREM2[33]. This could imply that the variant is directly involved in the pathogenesis of LOAD. In summary, these results demonstrate that the effect of genetic drivers on major brain-linked diseases like LOAD can be readily detected in serum to both inform on the causal relationship and the directionality of the risk mediating effect. This would also suggest that serum may be an accessible proxy for microglia function, and cognition.

Variable levels of the cell adhesion protein SVEP1 are associated with variants located at chromosomes 1 and 9 (Supplementary Data 1, Fig. 5a and Supplementary Fig. 5). Genetic associations to SVEP1 levels at chromosome 9 include the low-frequency missense variant rs111245230 in SVEP1 (NP_699197.3: pD2702G) (Fig. 5b), which was recently linked to coronary heart disease (CHD), blood pressure, and type-2-diabetes (T2D)[34]. In total, we found four conditionally independent missense mutations in *SVEP1* that were

**Table 1 Selected examples of exome array variants affecting serum protein levels and complex disease.**

| Disease class | Disease trait | PMID or database | pQTL | GWAS lead SNP(s)[a] | Function pSNP[b] | Mapped GWAS locus[c] | #Proteins affected | Example of *cis* and/or *trans* affected proteins[d] |
|---|---|---|---|---|---|---|---|---|
| *Cardiovascular* | | | | | | | | |
| | CHD | 28714975 | rs12740374 | rs12740374 | 3'-UTR | CELSR2 | 8 | **C1QTNF1**, **IGFBP1** |
| | VTE | UKBB, 28373160 | rs2343596 | rs16873402, rs4602861 | Intron | ZFPM2 | 7 | **VEGFA**, **DKK1** |
| | Stroke | 26708676 | rs653178 | rs653178 | Intron | ATXN2 | 2 | **THPO**, **CXCL11** |
| *Metabolic* | | | | | | | | |
| | T2D | 22885922 | rs7202877 | rs7202877 | Intergenic | CTRB1 | 5 | CTRB1, **PRSS2**, **CPB1** |
| | VAT | 20935629 | rs9491696 | rs9491696 | Intron | RSPO3 | 1 | RSPO3 |
| | Triglyceride | 21386085 | rs2266788 | rs2266788 | 3'-UTR | APOA5 | 5 | APOA5, PCSK7, **ANGPTL3** |
| *CNS* | | | | | | | | |
| | LOAD | 21460840 | rs610932 | rs610932 | 3'-UTR | MS4A6A | 3 | **TREM2**, **GLTPD2** |
| | Parkinson | 21738487 | rs6599389 | rs6599389 | Intron | GAK | 1 | IDUA |
| | Schizophrenia | 25056061 | rs3617 | rs3617 | Q315K | ITIH3 | 8 | ITIH3, **JAKMIP3** |
| *Inflammatory* | | | | | | | | |
| | SLE, T1D | 26502338 | rs2304256 | rs2304256 | V362F | TYK2 | 2 | ICAM1, ICAM5 |
| | Crohn´s, IBD | 21102463 | rs11209026 | rs11209026 | R381Q | IL23R | 1 | IL23R |
| | AMD | 2355636 | rs10737680 | rs10737680 | Intron | CFH | 22 | CFH, CFHR1, **CFB** |
| *Cancer* | | | | | | | | |
| | Colorectal | 24836286 | rs2241714 | rs1800469 | I11M | TMEM91 | 3 | **B3GNT2**, TGFB1 |
| | Lung | 18978787 | rs3117582 | rs3117582 | Intron | *APOM* | 10 | *MICB*, **ISG15** |
| | Melanoma | 18488026 | rs910873 | rs910873 | Intron | PIGU | 1 | ASIP |

*CHD* coronary heart disease, *VTE* venous thromboembolism, *CKD* chronic kidney disease, *T2D* type 2 diabetes, *VAT* visceral adipose tissue, *LOAD* late-onset Alzheimer's disease, *SLE* systemic lupus erythematous, *IBD* inflammatory bowel disease, *AMD* age-related macular degeneration, *N/A* not applicable. All reported pQTL effects are genome-wide significant, using linear regression, at $P < 1.92 \times 10^{-10}$ (two-sided).
[a]Protein QTLs overlapping GWAS lead SNPs using the PhenoScanner database[23]. No SNP proxies were applied except when the lead pSNP was not in the query then we used the best proxy ($r^2 \geq 0.8$ between markers).
[b]The functional annotation of pQTLs was obtained from the PhenoScanner database[23].
[c]Reported causal candidates are from the GWAS Catalog and reaching genome-side significance ($P < 5 \times 10^{-8}$, two-sided)[71].
[d]The definition of *cis* vs. *trans* effects is somewhat arbitrary depending on the window size chosen across the protein gene in question. In this case, however, all affected proteins located at other chromosomes than the pQTL location were considered *trans* acting and are highlighted in bold letters. All significant pQTLs are listed in Supplementary Data 1 and 2, and the overlap with GWAS risk loci is summarized in Supplementary Data 4.

associated with SVEP1 serum levels (Supplementary Data 2). The CHD and T2D risk allele (C) of rs111245230 were associated with elevated levels of SVEP1, and SVEP1 levels were elevated in T2D patients (OR = 1.20, $8 \times 10^{-5}$) and predictive of incident CHD (OR = 1.21, $8 \times 10^{-9}$) (Fig. 5c). Furthermore, high SVEP1 levels were positively associated with systolic blood pressure ($\beta = 0.266$, $P = 4 \times 10^{-9}$) (Fig. 5c), but not with diastolic blood pressure ($\beta = 0.028$, $P = 0.535$) (Fig. 5c). Consistently, higher serum levels of SVEP1 were associated with increased mortality post-incident CHD in the AGES-RS (HR = 1.28, $P = 3 \times 10^{-9}$) (Fig. 5d). The variants at chromosome 1 linked to SVEP1 levels (Fig. 5a), have not previously been linked to any disease. Given the currently available GWAS summary statistics, a two-sample MR analysis using *cis*-variants on chromosome 9 for SVEP1 as instruments and GWAS associations for T2D[35] support a causal relationship of SVEP1 with the disease ($P = 5.7 \times 10^{-6}$) (Fig. 5e, Supplementary Data 5), but not with CHD[36] or systolic blood pressure[37] ($P > 0.05$) (Supplementary Data 5). Our data triangulation and causal tests integrating genetics, serum protein levels, and disease(s), indicate that SVEP1 may be a therapeutic target for T2D.

The ILMN exome array contains several tags related to previous GWAS findings[38], including many risk loci for cancer. For example, 21 loci have been associated with melanoma[39] and 50 loci with colorectal cancer[40]. The exome array variant rs910873 located in an intron of the GPI transamidase gene *PIGU* was previously linked to melanoma risk[41]. The reported candidate gene *PIGU* is the gene most proximal to the lead SNP rs910873 and maybe a novel candidate gene involved in melanoma. However, a more biologically relevant candidate is the agouti-signaling protein (*ASIP*) gene that is located 314 kb downstream

of the lead SNP rs910873. *ASIP* is a competitive inhibitor of MC1R[42] and is thus strongly biologically implicated in melanoma risk[43]. We found that the melanoma risk allele for rs910873 was associated with elevated ASIP serum levels ($P = 5 \times 10^{-179}$) (Fig. 6a, Table 1), while the variant had no effect on other proteins measured with the current proteomic platform (Supplementary Data 1). Interestingly, the pQTL rs910873 is also an eQTL for *ASIP* gene expression in skin[44], showing the directionally consistent effect on the mRNA and protein. Importantly, we found that serum ASIP levels were supported as causally related to malignant melanoma ($P = 1.1 \times 10^{-17}$) using a two-sample MR analysis on the protein-to-outcome causal sequence of events (Fig. 6b, Supplementary Data 5). Our data point to the ASIP protein underlying the risk at rs910873, thus providing supportive evidence for the hypothesis that ASIP mediated inhibition of MC1R results in suppression of melanogenesis and increased risk of melanoma[45]. An additional example is the susceptibility variant rs1800469 for colorectal cancer[46], which is a proxy to the pQTL rs2241714 ($r^2 = 0.978$) (Table 1 and Fig. 6c). While the *TMEM91* gene was the reported candidate gene for the colorectal cancer risk at the rs1800469 (Table 1), we find that the risk variant affected three proteins in either *cis* (B3GNT8 and TGFB1) or *trans* (B3GNT2) (Fig. 6c, d). Intriguingly, all three proteins have previously been implicated in colorectal cancer[47–49]. Due to a lack of available and powered GWAS summary statistics data, we were unable to formally test the causality of these proteins to colorectal cancer. In conclusion, while we cannot rule out *PIGU* as a candidate gene for malignant melanoma, these findings point to an alternate, and possibly more biologically relevant candidate, ASIP.

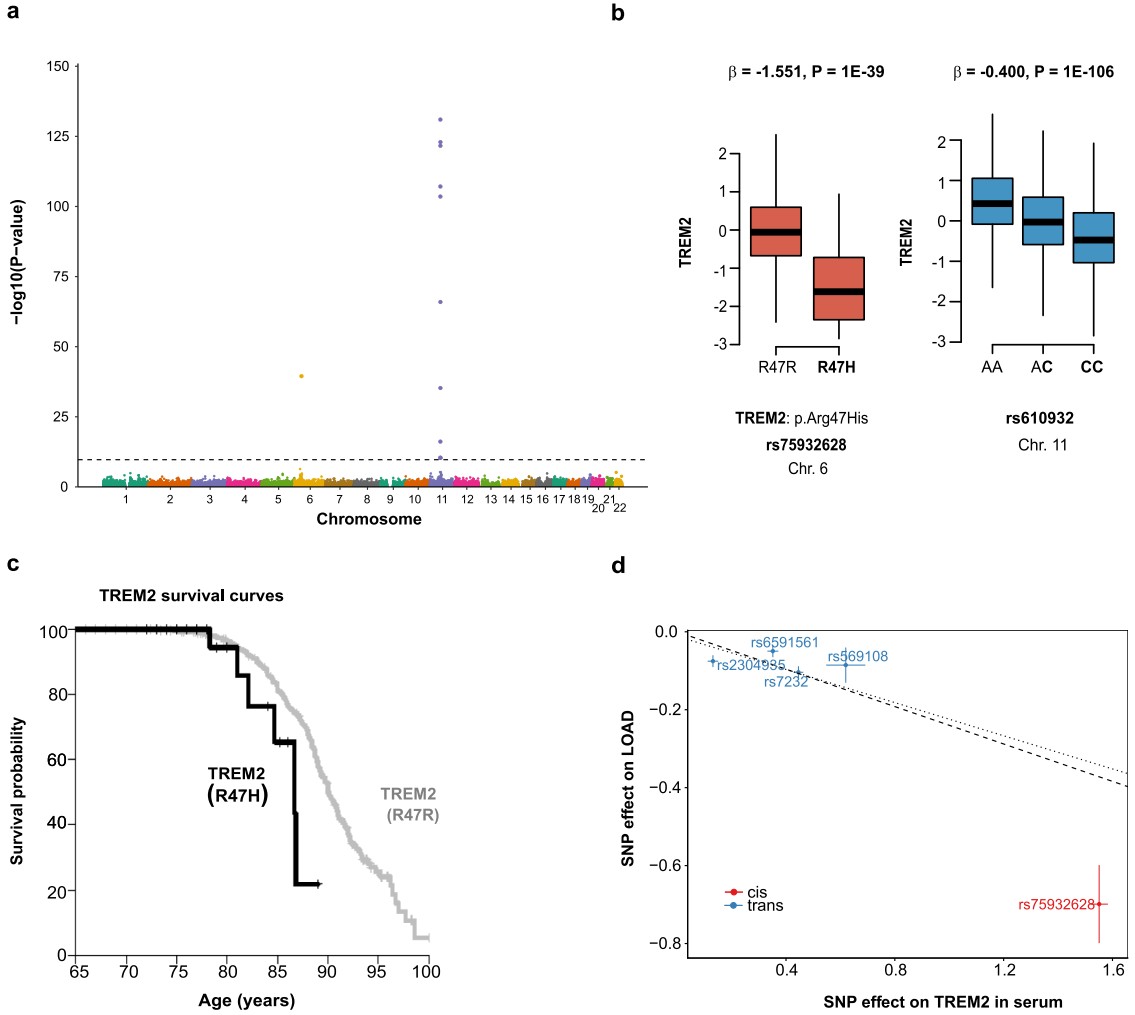

**Fig. 4 Effects of distinct risk loci for LOAD converge on the protein TREM2. a** The Manhattan plot highlights variants at two distinct chromosomes associated with serum TREM2 levels. Study-wide significant associations (linear regressions) at $P < 1.92 \times 10^{-10}$ (two-sided) are indicated by the horizontal line. The y-axis shows the $-(\log_{10})$ of the P-values for the association of each genetic variant on the exome array present along the x-axis. Variants at both chromosomes 6 and 11 associated with TREM2 have been independently linked to risk of LOAD including the rs75932628 (NP_061838.1: p.R47H) in TREM2 at chromosome 6 and the variant rs610932 at chromosome 11. **b** The boxplot to the left shows that carriers with the p.R47H mutation, which is linked to LOAD, are associated with low TREM2 levels. The boxplot on the right shows the *trans* effect of the well-established GWAS risk variant rs610932 for LOAD on TREM2 serum levels, where the LOAD risk allele C (highlighted in bold) is associated with lower levels of TREM2. The x-axis of each box plot shows the genotypes for the corresponding protein-associated SNP, while the y-axis denotes the Box–Cox transformed, age, and sex-adjusted serum protein levels. Box plots indicate median (middle line), 25th, 75th percentile (box), and 5th and 95th percentile (whiskers). The P-values (two-sided) shown at the top of each plot come from linear regression analysis. **c** TREM2p.R47H carriers demonstrated lower survival probability post-incident LOAD compared to TREM2p.R47R carriers ($P = 0.04$, two-sided). The vertical ticks correspond to individuals lost to follow-up. **d** Scatterplot for the TREM2 protein supported as having a causal effect on LOAD in a two-sample MR analysis. The figure demonstrates the estimated effects of the respective *cis*- and *trans*-acting genetic instruments on the serum TREM2 levels in AGES-RS (x-axis) and risk of LOAD through a GWAS by Kunkle et al.[32] (y-axis), using 21,982 LOAD cases and 41,944 controls. Each data point displays the estimated effect as beta coefficient = log(odds ratio), along with 95% confidence intervals for the SNP effect on disease (vertical lines) or SNP effect on the protein (horizontal lines). The broken line indicates the inverse variance weighted causal estimate ($\beta = -0.240$, SE = 0.059, $P = 5.3 \times 10^{-5}$, two-sided), while the dotted line shows the MR-Egger regression (see Supplementary Data 5 for more details).

We outlined the construction of the serum protein network in our previous report and identified common genetic variants underlying the network structure[10]. This included a targeted study of the effects of common *cis* and *cis*-to-*trans* acting variants on levels of serum proteins. Previously, we discovered that 80% of *cis* pQTL effects and 74% of *trans* pQTL effects were replicated across different populations and proteomics platforms measuring common variants[10]. We estimated the novelty of pQTL findings reported in the present study at both SNP–protein and locus-protein levels (see Supplementary Note 1 for details). In brief, using all conditionally independent study-wide significant

associations (Supplementary Data 2) and a linkage disequilibrium (LD) threshold of $r^2 < 0.5$ for novel associations, the current study's SNP–protein associations are 76.8% novel compared to Emilsson et al.[10], 75.5% novel compared to Sun et al.[11], and 59.3% novel compared to all published pQTL studies (Supplementary Fig. 6A, Supplementary Data 6 and Supplementary Note 1). Similarly, in comparison to our companion GWAS paper[50] and using the same LD threshold for novel associations, we find that 48.4% were exome-array-specific (Supplementary Fig. 6B, Supplementary Note 1). By combining all unique and common SNP–protein signatures from both companion studies,

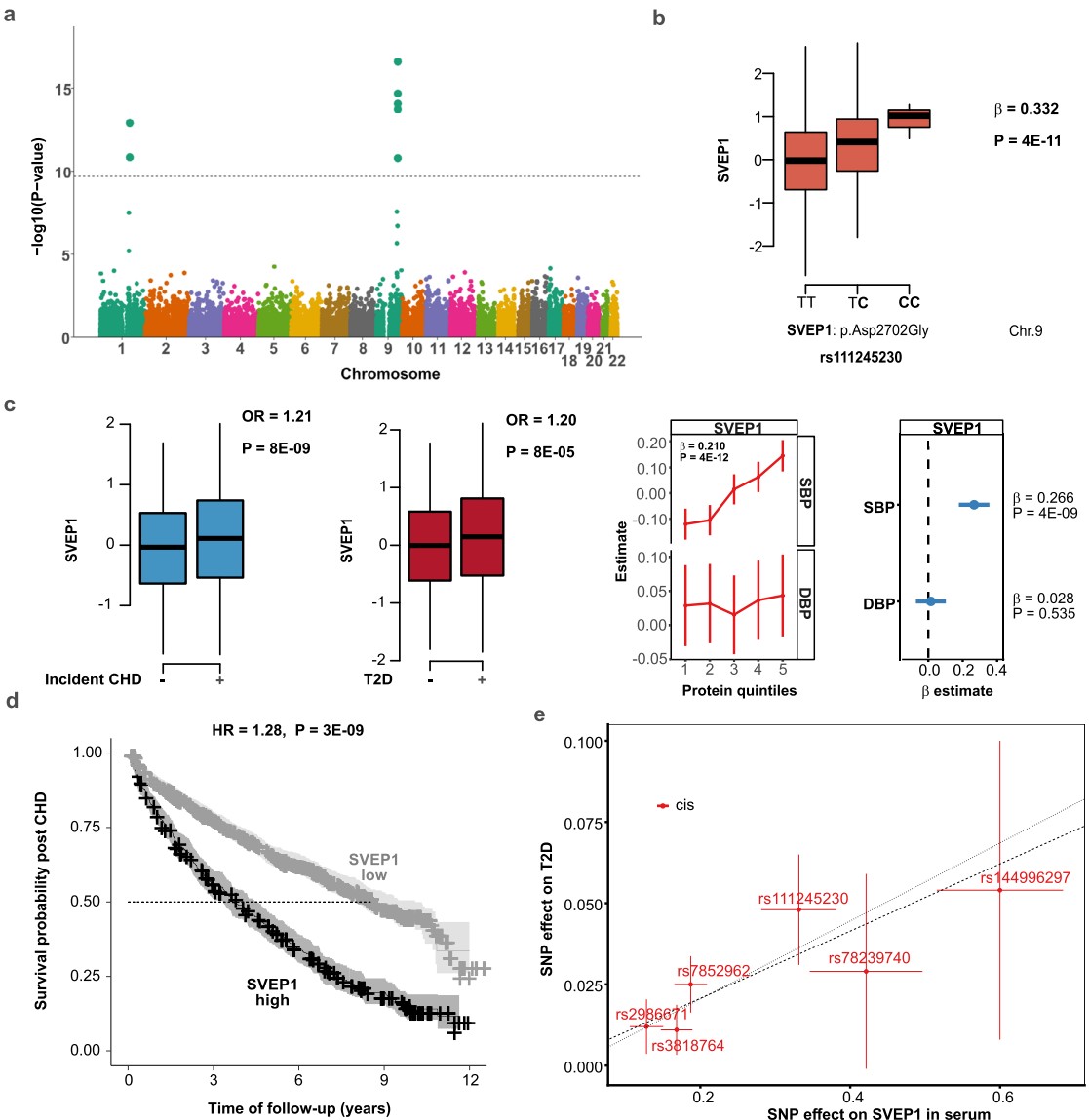

**Fig. 5 Variants affecting SVEP1 levels are associated with CHD, blood pressure, and T2D. a** The Manhattan plot reveals variants at chromosomes 1 and 9 associated with serum SVEP1 levels. Study-wide significant associations (linear regression, $P < 1.92 \times 10^{-10}$, two-sided) are indicated by the horizontal line. The $y$-axis shows the $-(\log_{10})$ of the $P$-values for the association of each genetic variant on the exome array present along the $x$-axis. **b** One of the variants associated with SVEP1 levels and underlying the peak at chromosome 9 is the low-frequency CHD risk variant rs111245230 (NP_699197.3: pD2702G). The CHD risk allele C (highlighted in bold) is associated with increased serum SVEP1 levels. The $x$-axis of the box plot shows the genotypes for the protein-associated SNP, while the $y$-axis denotes the Box–Cox transformed, age, and sex-adjusted serum protein levels. The $P$-value (two-sided) shown at the top of the plot is derived from linear regression analysis. Box plots indicate median (middle line), 25th, 75th percentile (box), and 5th and 95th percentile (whiskers). **c** Serum levels of SVEP1 were associated with incident CHD ($P = 8 \times 10^{-9}$) and T2D ($P = 8 \times 10^{-5}$). The $P$-values (two-sided) at the top of each boxplot for CHD and T2D come from logistic regression. The comparison of protein quintiles of the SVEP1 levels in serum with systolic (SBP) or diastolic (DBP) show a significant positive correlation with SBP ($\beta = 0.210$, $P = 4 \times 10^{-12}$, two-sided) but not with DBP ($P > 0.05$, two-sided). The relationship between the top and bottom quintiles of serum SVEP1 levels and blood pressure is depicted in the right-most panel. The $x$-axis of the box plots shows the health status of individuals, while the $y$-axis denotes the Box–Cox transformed, age, and sex-adjusted serum protein levels. Box plots indicate median (middle line), 25th, 75th percentile (box), and 5th and 95th percentile (whiskers). **d** Consistent with the directionality of the effects described above, we find that elevated levels of SVEP1 were associated with higher rates of mortality post-incident CHD. The Kaplan–Meier plot calculates the hazard ratio (HR) by comparing the 75th and 25th percentiles of SVEP1 serum levels. The vertical ticks correspond to individuals lost to follow-up while the shaded areas indicated the 95% confidence intervals. The $P$-value (two-sided) and HR are shown at the top of the plot. **e** Scatterplot for the SVEP1 protein supported as having a causal effect on T2D in a two-sample MR analysis. The figure demonstrates the SNP effect on serum SVEP1 levels ($x$-axis) and T2D from a GWAS in Europeans[35] ($y$-axis), with 74,124 T2D patients and 824,006 controls. Each center data point displays the estimated effect as beta coefficient = log(odds ratio), along with 95% confidence intervals for the SNP effect on disease (vertical lines) or SNP effect on the protein (horizontal lines). The broken line indicates the inverse variance weighted causal estimate ($\beta = 0.104$, SE = 0.023, $P = 5.7 \times 10^{-6}$, two-sided), while the dotted line demonstrates the MR–Egger regression (see Supplementary Data 5).

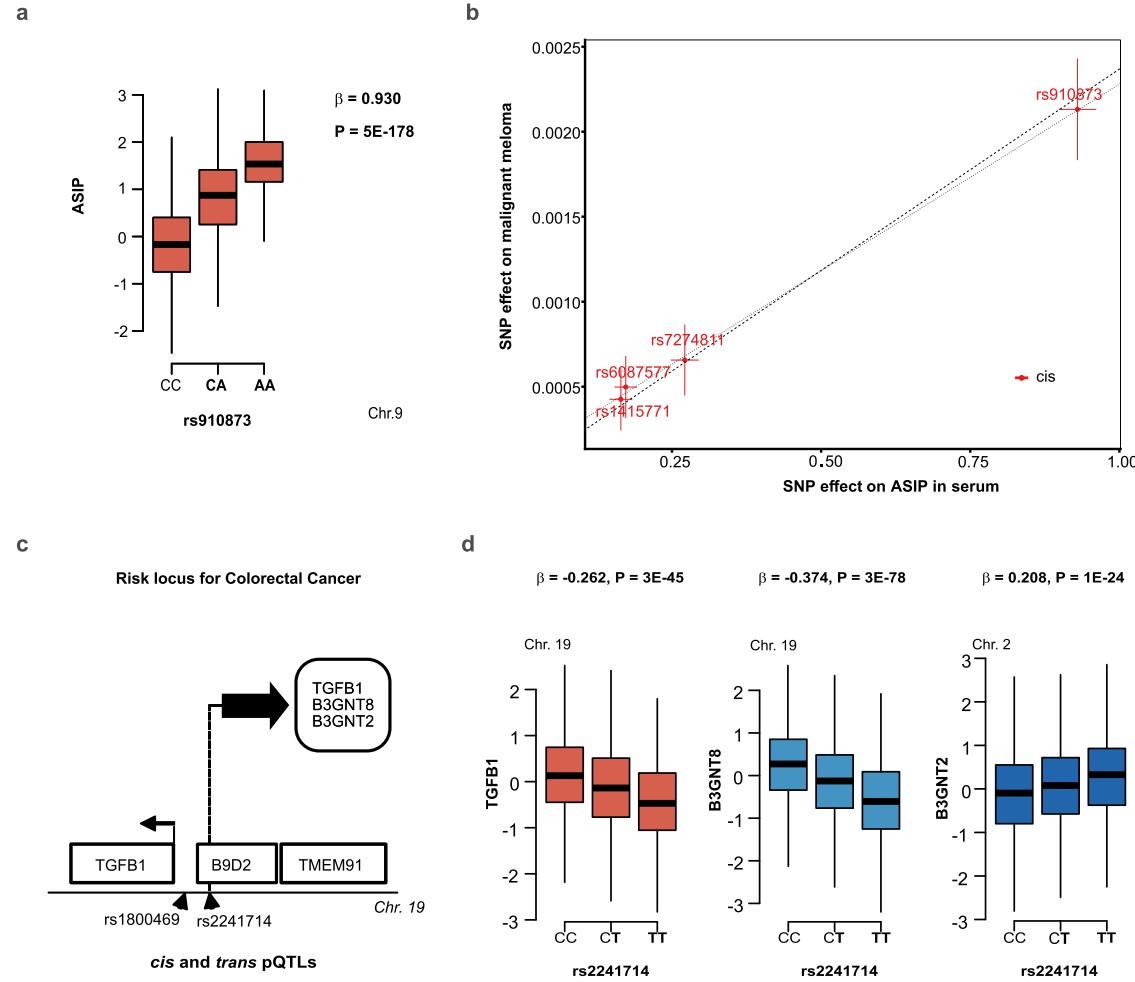

**Fig. 6 Proteins associated with malignant melanoma and colorectal cancer. a** The melanoma risk allele A (highlighted in bold) for the variant rs910873 is associated with high serum levels of ASIP. The *x*-axis of the box plot shows the genotypes for the protein-associated SNP, while the *y*-axis denotes the Box–Cox transformed, age, and sex-adjusted serum protein levels. Box plots indicate median (middle line), 25th, 75th percentile (box), and 5th and 95th percentile (whiskers). The *P*-value (two-sided) shown at the top of the plot is from linear regression analysis. **b** Scatterplot for the ASIP protein supported as having a causal effect on malignant melanoma in a two-sample MR analysis. The figure demonstrates the estimated effects of the respective genetic instruments on the serum ASIP levels in AGES-RS (*x*-axis) and risk of melanoma in GWAS by UK biobank data (UKB-b-12915)[67] (*y*-axis), that included 3598 melanoma cases and 459,335 controls. Each center data point displays the estimated effect as beta coefficient = log(odds ratio), along with 95% confidence intervals for the SNP effect on disease (vertical lines) or SNP effect on the protein (horizontal lines). The broken line indicates the inverse variance weighted causal estimate ($\beta = 0.0024$, SE = 0.0003, $P = 1.1 \times 10^{-17}$, two-sided), while the dotted line shows the MR-Egger regression (see Supplementary Data 5). **c** The pQTL rs2241714 is a proxy for colorectal cancer-associated variant rs1800469 ($r^2 = 0.978$) (Supplementary Data 2), located within the gene *B9D2* and proximal to *TMEM91* which is the reported candidate gene at this locus (see Table 1). The gene encoding TGFB1, a protein linked to rs2241714 in *cis*, is also nearby. **d** The variant rs2241714 (and rs1800469) is associated with the serum proteins TGFB1 (in *cis*), B3GNT8 (in *cis*), and B3GNT2 (in *trans*). The *P*-values (two-sided) shown at the top of each plot are from linear regression analysis. The *x*-axis of each box plot shows the genotypes for the corresponding protein-associated SNP, while the *y*-axis denotes the Box–Cox transformed, age, and sex-adjusted serum protein levels. Box plots indicate median (middle line), 25th, 75th percentile (box), and 5th and 95th percentile (whiskers). The chromosomes indicated at the top of each graph correspond to the location of the gene that encodes the protein of interest.

we obtain 6362 SNP–protein associations, of which 60.0% (at LD threshold of $r^2 < 0.5$) are novel when compared to external pQTL datasets (Supplementary Note 1, Supplementary Fig. 6C). Finally, when estimating novelty at the locus–protein level, we find that 321 out of 881 loci and 762 out of 3103 locus–protein associations identified in the current study are novel compared to our companion paper[50] (Supplementary Data 7, Supplementary Note 1). When the two companion studies were combined, they yielded 404 new loci and 1950 new locus-protein associations, which were not found in previous pQTL publications (Supplementary Data 6, Supplementary Data 7, and Supplementary Note 1).

We report here that many of the measured serum proteins under genetic control share genetics with a variety of clinical features, including major diseases arising from various body tissues. This is in line with a recent population-scale survey of human-induced pluripotent stem cells, demonstrating that pQTLs are 1.93-fold enriched in disease risk variants compared to a 1.36-fold enrichment for eQTLs[12], underscoring the added value in pQTL mapping. We reaffirm widespread associations between genetic variants and their cognate proteins as well as distant *trans*-acting effects on serum proteins and demonstrate that many proteins are often involved in mediating the biological

effect of a single causal variant affecting complex disease. Protein coding variants may cause technical artifacts in both affinity proteomics and mass spectrometry[51,52]. Systematic conditional and colocalization studies have shown, however, that pQTLs powered by common missense variants being artefactual are not a common event using the aptamer-based technology[11,53], however, given the enrichment of missense variants in the present study, it may occur in some cases.

We note that with the ever-increasing availability of large-scale omics data aligned with the human genome, cross-referencing different datasets can result in findings that occurred by sheer chance. Hence, a systematic colocalization analysis has been proposed for identifying shared causal variants between intermediate traits and disease endpoints[54]. This is, however, not feasible for the application of the exome array given its sparse genomic coverage. Instead, multi-omics data triangulation to infer consistency in directionality, the approach used in the present study, can enhance confidence in the causal call and offer insights and guidelines for experimental follow-up studies. In fact, the causal calls for TREM2 (LOAD), SVEP1 (T2D), and ASIP (malignant melanoma) were validated, using a two-sample MR analysis. These analyses found no evidence of horizontal pleiotropy (Supplementary Data 5), nor did they demonstrate that the causal estimates were dependent on a single genetic instrument (Supplementary Fig. 7). We previously asserted that serum proteins are intimately connected to and may mediate global homeostasis[10]. The accumulated data show that serum proteins are under strong genetic control and closely associated with diseases of different aetiologies, which in turn suggests that serum proteins may be significant mediators of systemic homeostasis in human health and disease.

## Methods

**Study population.** The AGES-RS[55] was approved by the NBC in Iceland (approval number VSN-00-063), and by the National Institute on Aging Intramural Institutional Review Board, and the Data Protection Authority in Iceland. AGES-RS is a single-center prospective population-based study of deeply phenotyped subjects (5764, mean age 75 ± 6 years) and survivors of the 40-year-long prospective Reykjavik study ($N \sim 18,000$), an epidemiologic study aimed to understand aging in the context of gene/environment interaction by focusing on four biologic systems: vascular, neurocognitive (including sensory), musculoskeletal, and body composition/metabolism. Descriptive statistics of this cohort as well as the detailed definition of the various disease endpoints and relevant phenotypes measured have been published[10,55].

**Genotyping platform.** Study samples were processed on the exome-wide genotyping array Illumina HumanExome BeadChip v1.0 (San Diego, CA, USA) for all AGES-RS participants at the University of Texas Health Science Center at Houston genotyping center as previously described[56]. The exome array was enriched for exonic variants selected from over 12,000 individual exome and whole-genome sequences from different study populations[38] and includes as well tags for previously described GWAS hits, ancestry informative markers, mitochondrial SNPs, and human leukocyte antigen tags[38]. A total of 244,883 variants were included on the exome array. Genotype call and quality control filters including call rate, heterozygosity, sex discordance, and principal component analysis outliers were performed as previously described[2,21]. Variants with call rate <90% or with Hardy–Weinberg $P$ values $< 1 \times 10^{-7}$ were removed from the study. Totally, 76,891 variants were detected in at least one individual of the AGES-RS cohort. Of these variants, 54,469 had a MAF > 0.001 and were examined for association against each of the 4782 human serum protein measurements (see below).

**Protein measurements.** Each protein has its own detection reagent selected from chemically modified DNA libraries, referred to as SOMAmers[57]. The design and quality control of the SOMApanel platform's custom version to include proteins known or predicted to be present in the extracellular milieu have been described in detail elsewhere[10]. Briefly, though, the aptamer-based platform measures 5034 protein analytes in a single serum sample, of which 4782 SOMAmers bind specifically to 4137 human proteins (some proteins are identified by more than one aptamer) and 250 SOMAmers that recognize non-human targets (47 non-human vertebrate proteins and 203 targeting human pathogens)[10]. Consistent target specificity across the platform was indicated by direct (through mass spectrometry) and/or indirect validation of the SOMAmers[10]. Both sample selection and sample

processing for protein measurements were randomized, and all samples were run as a single set to prevent batch or time of processing biases.

**Statistical analysis.** Prior to the analysis of the proteins measurements, we applied a Box–Cox transformation on all proteins to improve normality, symmetry and to maintain all protein variables on a similar scale[58]. In the association analysis, we obtained residuals after controlling for sex, age, potential population stratification using principal component (PCs) analysis[59], and for all single-variant associations to serum proteins tested under an additive genetic model applying linear regression analysis (protein ∼SNP + age + sex + PC1 + PC2 + ….PC5). We report both variants to protein associations at $P < 1 \times 10^{-6}$ for suggestive evidence and Bonferroni correction for multiple comparisons by adjusting for the 54,469 variants and 4782 human protein analytes where single variant associations with $P < 1.92 \times 10^{-10}$ were considered study-wide significant (Supplementary Data 1). $P$-values corresponding to the estimated effect size and standard errors of the genotypes were recalculated to increase accuracy. Independent genetic signals were found through a stepwise conditional and joint association analysis for each protein analyte separately with the GCTA-COJO (v1.92.4beta2) software[60,61]. We conditioned on the current lead variant listed in Supplementary Data 1, defined as the variant with the lowest $P$-value, and then kept track of any new variants that were not in LD (the default GCTA-COJO option $r^2 < 0.9$ for co-linearity) with previously chosen lead variants and reported findings at $P < 1 \times 10^{-6}$ (Supplementary Data 2). In the joint model, all conditionally significant SNPs for each protein analyte were combined in the regression model.

Supplementary Data 3 summarizes, through the use of VEP (v104.0)[24,25], various pathogenicity prediction scores for all independent study-wide significant pQTLs in Supplementary Data 2, including the Likelihood Ratio Test[62], Variant Effect Scoring Tool[63], MutationAssessor[64], and MutationTaster[65].

To test whether the percentage of secreted proteins among pQTLs is equal to the percentage of secreted proteins among non-pQTLs, 10,000 permutations were performed to obtain the empirical distribution of the $\chi^2$ test of equality of proportions. Our null and alternate hypotheses were:

$H_0$: $P(\text{pQTL} \mid \text{Secreted}) = P(\text{pQTL} \mid \text{Not Secreted})$ and $H_1$: $P(\text{pQTL} \mid \text{Secreted}) > P(\text{pQTL} \mid \text{Not Secreted})$ The test statistics calculated from our data was compared to the quantiles of this distribution to obtain $P(\text{Data} \mid H_0)$ (Supplementary Fig. 1).

We applied the "TwoSampleMR" R package[66] to perform a two-sample MR analysis to test for causal associations between protein and outcome (protein-to-outcome). For different outcomes, we used GWAS associations for LOAD in Europeans[32], malignant melanoma in European individuals from the UK biobank data (UKB-b-12915)[67], T2D in Europeans[35], CHD in Europeans[36], and systolic blood pressure in Europeans[37]. Genetic variants (SNPs) associated with serum protein levels at a genome-wide significant threshold ($P < 5 \times 10^{-8}$) identified in the AGES-RS dataset and filtered to only include uncorrelated variants ($r^2 < 0.2$) were used as instruments. More to the point, genetic instruments within the cis window for each aptamer were then clumped such that variants in high LD ($r^2 \geq 0.2$) within a 10 Mb region were combined using the LD structure of the AGES-RS population. The inverse variance weighted (IVW) method[68] was used for the MR analysis, with $P$-values $< 0.05$ considered significant. For sensitivity analyses, we used the intercept term from MR Egger regression[69] to determine whether there was evidence of horizontal pleiotropy, and Cochran's $Q$-statistic[70] to evaluate heterogeneity of genetic instruments. A leave-one-out analysis was also performed to see the effect of individual SNPs on the causal estimate. A bi-directional MR analysis was also attempted but not concluded as there were no overlapping SNPs between the exome GWAS and the GWAS' for LOAD, malignant melanoma, and T2D after we had filtered them for significant associations.

For the associations of individual proteins to different phenotypic measures, we used linear or logistic regression or Cox proportional hazards regression, depending on the outcome being continuous, binary, or a time to an event. Given the consistency in terms of sample handling including time from blood draw to processing (between 9 and 11 am), same personnel handling all specimens, and the ethnic homogeneity of the population we adjusted only for age and sex in all our regression analyses. All statistical analysis was performed using R version 3.6.0 (R Foundation for Statistical Computing, Vienna, Austria) and RStudio (v1.1.456).

We compared our pQTL results to 19 previously published proteogenomic studies (Supplementary Data 5), including the protein GWAS in the INTERVAL study[11], and we previously reported genetic analysis of 3219 AGES-RS cohort participants[10]. In the previous proteogenomic analysis of AGES-RS participants, one cis variant was reported per protein using a locus-wide significance threshold, as well as cis-to-trans variants at a Bonferroni corrected significance threshold. Due to these differences in reporting criteria, we only considered the associations in previous AGES-RS results that met the current study-wide $P$-value threshold. For all other studies, we retained the pQTLs at the reported significance threshold. In addition, we performed a lookup of all independent pQTLs from the current study available in summary statistics from the INTERVAL study, considering them known if they reached a study-wide significance in their data. We calculated the LD structure between the reported significant variants for all studies, using 1000 Genomes v3 EUR samples, but using AGES-RS data when comparing to previously reported AGES-RS results. We considered variants in LD at $r^2 > 0.5$ to represent

the same signal across studies. The comparison was performed on protein level, by matching the reported Entrez gene symbol from each study.

**Reporting summary**. Further information on research design is available in the Nature Research Reporting Summary linked to this article.

## Data availability

The custom-design Novartis SOMAscan is available through a collaboration agreement with the Novartis Institutes for BioMedical Research (lori.jennings@novartis.com). Data from the AGES-RS study are available through collaboration (AGES_data_request@hjarta.is) under a data usage agreement with the IHA. All-access to data is controlled via the use of a subject-signed informed consent authorization. The time it takes to respond to requests varies depending on the nature and circumstances of the request, but it will not exceed 14 working days. All data supporting the conclusions of the paper are presented in the main text and freely available as a supplement to this manuscript (Supplementary Information and Supplementary Data).

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

## Acknowledgements

The authors acknowledge the contribution of the Icelandic Heart Association (IHA) staff to AGES-RS, as well as the involvement of all study participants. The National Institute on Aging (NIA) contracts N01-AG-12100 and HHSN271201200022C for V.G. financed the study. V.G. received funding from the NIA (1R01AG065596), and IHA received a grant from Althingi (the Icelandic Parliament). The Icelandic Research Fund (IRF) funded V.E. and Va.G. with grants 195761-051, 184845-053, and 206692-051, while Va.G. received a postdoctoral research grant from the University of Iceland Research Fund. M.A.K. was funded by Open Targets and by the Wellcome Trust Grant 206194.

## Author contributions

V.E and Vi.G. designed the study. A.G., Va.G., E.F.G., T.J, B.G.J., J.R.L., M.A.K., M.I., J.R.S., T.A., and V.E. performed data analysis. L.L.J., L.J.L., J.H.L., and N.M.M. provided expertise in a variety of areas, including proteomics data and function, and contributed to the discussion. V.E. and Vi.G. supervised the project. V.E. wrote the first draft of the paper, with all coauthors contributing to data interpretation, paper editing, and revision.

## Competing interests

The study was supported by the Novartis Institute for Biomedical Research, and protein measurements for the AGES-RS cohort were performed at SomaLogic. J.R.L. and L.L.J. are employees and stockholders of Novartis. The remaining authors declare no competing interests.
