## [Peer Review File · Nature Communications]

Coding and regulatory variants associated with serum protein levels and diseaseReviewers' Comments:

Reviewer #1:

Remarks to the Author:

The present manuscript describes an exome-wide association study with levels of 4782 blood circulating proteins that were measured on the SOMAscan platform in 5,457 individuals of the AGES Reykjavik cohort.

The paper reports 5,553 variants affecting levels of 1931 serum proteins and finds overlaps with genetic loci for hundreds of complex disease traits.

It is not clear how much of this work overlaps with a previously reported genome-wide association study on the same dataset and that reported similar results (Emilsson et al., Science, 2018). Probably a lot.

The authors should report and discuss the overlap/difference between the 55,932 low-frequency and common exome-array variants used here from Illumina HumanExome BeadChip exome array and the variants imputed from the Illumina 370CNV BeadChip array in the 2018 Science paper.

How many of the 55,932 are in the imputed 370CNV data set and how well do the overlapping genotyped and imputed variant calls agree? How many are in strong LD? How many of the variants that are not covered by the imputed 370CNV data are associated with protein levels?

The authors acknowledge that a systematic colocalization analysis would be of interest for causality tests between intermediate traits and disease endpoints but argue that this is not feasible for application of the exome array given its sparse genomic coverage. However, this could be done if the genotype data from both arrays would be integrated and jointly analyzed.

The authors state that they "highlight how triangulation of data from different sources can link genetics, protein levels and disease(s)". I would have expected a systematic Mendelian Randomization study here, not just a report of the overlap between pQTLs and GWAS hits.

Epitope-changing variants can affect aptamer binding and lead to pQTL associations that are not reflected in changed protein levels. The authors write "systematic conditional and colocalization analyses in causality testing using the aptamer-based technology have shown that pQTLs driven by common missense variants being artefactual is an unlikely event [refs 11&45]". I disagree with the generality of this statement. Refs 11&45 acknowledge the existence of epitope effects, but they don't conclude that they are unlikely events – they are unlikely if the variant colocalizes with a GWAS signal on another trait, which is what the authors do not provide here. In addition, coding variants that can lead to epitope effects are enriched on the Exome array.

In summary, while I am generally positive about any new proteomics GWAS, I am a bit hesitant in recommending this paper for publication as it stands. The paper would be much stronger if it integrated the full genotyping data set and then used this as a basis to test for signal colocalization and Mendelian randomization.

Reviewer #2:

Remarks to the Author:

This study examined the overlap and association of genetic disease signatures with circulating serum proteins using the AGES Reykjavik cohort. Overall, this study was well designed, and the manuscript was well-written and interesting. The study makes nice use of data integration to provide supporting evidence for the conclusions that are drawn. The results of the analysis lend support to a growing body of evidence that circulating proteins not only act as biomarkers but in some cases mediate or cause disease. The results are suggestive of the possibility of targeted monitoring of serum protein content for people with certain risk alleles. However, despite the strengths of this article I do have some minor comments.

1. Lines 100-101: A Fisher's exact test was used, which I acknowledge is fairly common in enrichment tests. However, the independence assumption is not very likely to hold in this case, especially given the later claims of pleiotropy. A permutation or bootstrap test would be likely be a better choice.

2. Lines 104-106: discuss overlap of pQTLs with known GWAS loci, but the reasoning here seems a bit circular. Isn't the HumanExome BeadChip enriched for known GWAS loci? My point is not that the results are not valid or interesting, simply that this overlap in and of itself may not be.

3. Lines 111-112: suggest "that greater regulatory pleiotropy of pQTLs is associated with greater chance of disease trait pleiotropy". However, the phenotypes from PhenoScanner are not all independent (e.g. some are merely different ways of diagnosing the same condition) and I'm not sure about the independence of the serum proteins either. This makes correlation a tricky measure and it is not clear to me if this was taken into account. The Spearman correlation of 0.22 is not necessarily so high as to provide a lot of buffer on this front.

4. Lines 182-183: states "60% of the serum proteome that is under genetic controls shares genetics with reported clinical traits". However, the entire serum proteome is not measured, and it is not clear what proportion is under genetic control from these results. I would just reword this a bit.

Reviewer #3:

Remarks to the Author:

In their manuscript titled "Human serum proteome profoundly overlaps with genetic signatures of disease", Emilsson et al. studied associations of 55932 low-frequency and common exonic variants with 4782 protein measures in serum samples of 5457 individuals from AGES Reykjavik study. The authors identified 5553 variants associating with levels of 1931 serum proteins, and they further characterized the overlap of the genetic associations between proteins and human diseases.

The manuscript has multiple positive aspects. It provides a larger sample size and a larger number of proteins studied than the previous reports describing genome-wide associations of human blood proteome[1,2], it is well written, and it would be of interest to researchers in multiple fields of science. Unfortunately, the reviewer is not convinced if the manuscript fulfills the novelty criteria that can be expected from a high-quality journal like Nature Communications, as the data has been published as part of the previous work by the authors[3]. Although the focus of the previous paper is in describing serum protein networks and the description of the genetic associations is somewhat in the background, the same proteome and exome data in 5457 participants of AGES have been analyzed. The high overlap between the two papers is highlighted by the fact that the text in the Methods paragraph is partially identical to the text in Supplemental Methods of the previous publication[3] (lines 264-269, for example).

Also, the following issues should be addressed:

1. The association signals should be better characterized to identify the number of independent associations within a genomic region. For example, in lines 147-148, the authors describe that they "found eight different missense mutations in SVEP1" – to determine if these missense variants associate with SVEP1 level independently from each other, conditional analyses should be performed. One option to do this is with GCTA software using summary-level data and linkage disequilibrium (LD) matrix[4,5]. Additionally, variant annotation tools, such as SIFT, PolyPhen2, or MutationTaster (all implemented in ANNOVAR[6]), could be used to gain insights about the consequences of the variants associated with protein levels and to rank adjacent variants to identify the potentially causal ones.
2. The results presented in the manuscript are based on a single population. According to the reporting summary, independent replication was not possible due to a lack of suitable data in other cohorts. The authors should investigate if the loci reported here were reported in the previous studies (for the overlapping proteins)[1,2]. Replication of the known genetic associations provides assurance also for the novel associations.
3. The authors mention the "triangulation of data" on a few occasions. Still, they have not performed Mendelian randomization analyses, which would help to combine information from multiple association

tests to causal estimates between protein levels and human diseases[2,7]. It is probably beyond the scope of this manuscript to perform Mendelian randomization in all significant loci, but it would be beneficial to report causal estimates for the examples highlighted in the manuscript.

4. Population stratification is a major confounder in genetic association studies[8], and it can affect the validity of the association results even in populations considered to be relatively homogenous, such as the Icelanders[9]. The authors have not corrected the genetic association tests for population stratification.

Minor corrections/comments:

- The authors report the number of exome array variants (5553) associating with levels of serum proteins. It would be helpful to describe further the number of independent genomic regions associating with blood protein levels.
- Technically, "rs123456" is not a locus - the rs number is a unique identifier for a sequence variant in a locus (for example, rs2251219 is an identifier for allelic variation T/G/C in locus chr3:52550771). Usually, in genetic studies, "locus" refers to a genomic region larger than a single nucleotide. Please edit the incorrect expressions, such as "the locus rs2251219" on line 114.
- The methods section does not describe the software used for statistical analyses.
- In case the text above Figure 1b indicates gene names, the font should be italic.
- In Figures 3a and 4a, instead of showing Manhattan plots, it would be more useful to show the regional association plots of the significant association signals. These plots should include information about the LD structure. With the exome-wide data, the number of SNPs in the regional plots will be lower than in the case of genome-wide data, but they can still be very informative[10,11].
- In Figure 3d, it would be helpful to show the correlation coefficients on the plot.

References

1. Suhre, K. et al. Connecting genetic risk to disease end points through the human blood plasma proteome. *Nat. Commun.* 8, 14357 (2017).
2. Sun, B. B. et al. Genomic atlas of the human plasma proteome. *Nature* 558, 73–79 (2018).
3. Emilsson, V. et al. Co-regulatory networks of human serum proteins link genetics to disease. *Science* (80-.). 361, 769–773 (2018).
4. Yang, J. et al. Conditional and joint multiple-SNP analysis of GWAS summary statistics identifies additional variants influencing complex traits. *Nat. Genet.* 44, 369–75 (2012).
5. Yang, J., Lee, S. H., Goddard, M. E. & Visscher, P. M. GCTA: A tool for genome-wide complex trait analysis. *Am. J. Hum. Genet.* 88, 76–82 (2011).
6. Wang, K., Li, M. & Hakonarson, H. ANNOVAR: Functional annotation of genetic variants from high-throughput sequencing data. *Nucleic Acids Res.* 38, 1–7 (2010).
7. Hemani, G. et al. The MR-base platform supports systematic causal inference across the human phenome. *Elife* 7, 1–29 (2018).
8. Price, A. L. et al. Principal components analysis corrects for stratification in genome-wide association studies. *Nat. Genet.* 38, 904–909 (2006).
9. Helgason, A., Yngvadottir, B., Hrafnkelsson, B., Gulcher, J. & Stefansson, K. An Icelandic example of the impact of population structure on association studies. *Nat. Genet.* 37, 90–95 (2005).
10. Chang, J. et al. Exome-wide analyses identify low-frequency variant in CYP26B1 and additional coding variants associated with esophageal squamous cell carcinoma. *Nat. Genet.* 50, 338–343 (2018).
11. Natarajan, P. et al. Multiethnic Exome-Wide Association Study of Subclinical Atherosclerosis. *Circ. Cardiovasc. Genet.* 9, 511–520 (2016).

Response to Reviewers

We are pleased to submit our revised manuscript now entitled “*Coding and regulatory variants affect serum protein levels and common disease*” (NCOMMS-20-16689-T) for consideration to be published in *Nature Communications*. We are grateful for the reviewers’ thoughtful and insightful comments. As you will see in the accompanying documents, we have carefully considered, responded to, and addressed each of the comments point-by-point. We hope you will agree that the incorporation of these changes has resulted in a considerably stronger paper.

Please note that we have included two additional coauthors Karim MA (Sanger Institute at Cambridge, UK) and Jonsson BG (Icelandic Heart Association, Iceland), and moved the author Gudjonsson A to the joint first author position. All have contributed significantly to the additional analyses and drafting of the revised manuscript. We have added their affiliation and contribution information to the revised manuscript.

Responses below are provided in blue font. Text added to the revised manuscript has been italicized. Page and paragraph numbers listed below refer to the position of the new or modified text in the *clean* version of the revised manuscript (submitted along with a manuscript text file highlighting all changes using the track changes mode).

We would like to bring to your attention that we have also submitted another manuscript entitled “*A genome-wide association study of serum proteins reveals shared loci with common diseases*” to *Nature Communications*, which reports on the GWAS of all serum proteins measured across the AGES cohort with extensive colocalization analyses. We suggested to the editors, that the two papers be considered as companion studies. These two studies, we believe, are complementary and present the fullest telling to date of the genetics of serum proteins. We would like to highlight the novelty reported in each paper, as well as consider their combined novelty in comparison to what has been published by others to date. For example, when all independent SNP-to-protein associations in each study are compared using linkage disequilibrium (LD) of $r^2 > 0.50$ between markers, 57% (2296 of 4000 SNP-to-protein associations) are exome-array-specific, while 52% (2181 of 4227 SNP-to-protein associations) are GWAS-specific. This indicates that the two studies complement each other nicely as separate investigations. Using LD of $r^2 > 0.9$ to combine unique and common signatures from both studies, 65% (4135 of 6362 SNP-to-protein associations) are novel when compared to any external data sets, making this the most comprehensive proteogenomic analysis to date.

Reviewers' comments:

Reviewer #1

The present manuscript describes an exome-wide association study with levels of 4782 blood circulating proteins that were measured on the SOMAScan platform in 5,457 individuals of the AGES Reykjavik cohort. The paper reports 5,553 variants affecting levels of 1931 serum proteins and finds overlaps with genetic loci for hundreds of complex disease traits.

Comment 1. It is not clear how much of this work overlaps with a previously reported genome-wide association study on the same dataset and that reported similar results (Emilsson et al., Science, 2018). Probably a lot.

Response: We focused on building the serum protein networks for the previous paper¹ and explored the biology and genetic components underlying the various sub-networks. This was the central message, and the reporting of *cis* and *cis-to-trans* acting pQTLs was a by-product of that study, using a narrow *cis* window (a +/-150kb window across protein encoding genes). In addition, we focused on common variants (MAF \geq 0.05) genotyped in 3219 individuals in that study. Since then, the widely used exome array enriched for rare and low-frequency exonic variants was tested in 5343 AGES individuals, and is the focus of this work. Specifically, while 70% of the variants detected with the exome array are exonic and 59% of mapped pQTLs are exonic in this study, only 7% of the previously identified pQTLs were exonic¹. Also, the current study investigates the relationship between serum proteins and a variety of common diseases of diverse etiologies in contrast to the previous study¹, which focused predominantly on cardiometabolic diseases. This is to more fully explore the idea that the serum proteome mediates systemic homeostasis and is representative of the global disease status of individuals². However, we agree with the Reviewer (and Reviewer 3 below, sharing similar concerns) that it is important to report on the overlap between these two studies, which is detailed in our response to Reviewer 1's comment 2 below. There is certainly some overlap, but not a lot. To emphasize this, we have added a description to the main text (see our response to comment 2 below). Importantly, unlike the previous study, the current study investigates the relationship between circulating proteins and a more comprehensive set of measures of exome variants, revealing substantially novel associations (see responses below).

Comment 2. The authors should report and discuss the overlap/difference between the 55,932 low-frequency and common exome-array variants used here from Illumina HumanExome BeadChip exome array and the variants imputed from the Illumina 370CNV BeadChip array in the 2018 Science paper.

Response: We should point out that the entire association study was re-examined in order to address potential population stratification and identify independent SNP-to-protein associations (see below our responses to Reviewer 3's comments). The following direct comparison between both of these two platforms demonstrates that:

1. Using the P-value threshold $< 1 \times 10^{-6}$ for all independent associations in **Supplementary Table S2** and applying a Linkage Disequilibrium (LD) threshold $r^2 > 0.50$ for comparison (known associations), only 50 variants were explicitly overlapping between the two platforms. In the present analysis, 662 of the 5259 independent exome array variants showing

9659 associations with 2780 proteins were in LD ($r^2 > 0.50$) with any of the 1117 variants previously reported¹. In addition, only 12% of the 9659 genetic associations found in the present study, that is 1170 associations, were reported in our Science paper¹.

2. Using a study-wide significant P-value threshold of associations $< 1.92 \times 10^{-10}$ for independent associations in **Supplementary Tables S2**, there were 4092 study-wide significant independent associations between exome array variants and unique proteins (Entrez IDs), corresponding to 2019 SNPs and 2135 proteins. Of these, 49 SNPs are directly overlapping with the 1117 reported variants in the Science article¹. Using LD of $r^2 > 0.5$ between study specific variants as known associations (i.e. previously reported and thus not novel), 3161 SNP-to-protein associations are novel while 931 are known. In other words, only 22.8% were reported in the Science paper, implying that 77.2% of study-wide significant associations in the current study are novel.

To report the overlap (and distinction) between the two reports, we have added the following text on pages 8-9, lines 201-209:

*“We outlined the construction of the serum protein network in our previous report and identified common genetic variants underlying the network structure¹. This included a targeted study of the effects of common cis and cis-to-trans acting variants on levels of serum proteins. The comparison between that study and the current one using all independent study-wide significant associations (**Supplementary Table S2**) and linkage disequilibrium (LD) threshold of $r^2 > 0.50$ for known associations, shows that 77.2% of the current study's variant-to-protein associations are novel. Importantly, while 70% of the variants detected with the exome array are exonic and 59% of mapped pQTLs are exonic, only 7% of the identified pQTLs were exonic in our earlier report¹”*

Comment 3. How many of the 55,932 are in the imputed 370CNV data set and how well do the overlapping genotyped and imputed variant calls agree? How many are in strong LD? How many of the variants that are not covered by the imputed 370CNV data are associated with protein levels?

Response: When comparing variant calls across the two different genotyping platforms of the assayed and imputed variants, all exome-relevant variants were assayed, not imputed. The previously reported Illumina 370CNV platform (assayed and imputed through the 1000G reference panel) targets common variants ($MAF \geq 5\%$). In contrast, the exome array is enriched for rare ($MAF < 1\%$) and low-frequency ($1\% \geq MAF < 5\%$) variants. Because the genotype imputation of rare and low-frequency variants is of far lower quality than common variants³, and they are not present on the Illumina 370CNV platform, the comparison is not straightforward. With regards to the number of overlapping variants and variant-to-protein associations using a LD cutoff of $r^2 > 0.5$, we refer to our more comprehensive response to comment 2 by the Reviewer above.

Comment 4. The authors acknowledge that a systematic colocalization analysis would be of interest for causality tests between intermediate traits and disease endpoints but argue that this is

not feasible for application of the exome array given its sparse genomic coverage. However, this could be done if the genotype data from both arrays would be integrated and jointly analyzed.

Response: We note that the Illumina 370CNV platform was examined in 3219 individuals, while the exome array was applied to the entire AGES cohort, or 5343 individuals with protein measurements and detailed phenotype information. Regardless of the exome array's sparse genotype coverage, integrating disease relevant GWAS summary statistics data with protein QTLs is only possible if the variants affecting protein levels are common (MAF > 0.05). In other words, GWAS summary statistics data for disease (and expression QTLs) typically do not include information on rare and low-frequency variants found on the exome array. As a result, combining these two platforms will increase coverage in 57% of AGES subjects but will not allow for colocalization analysis in relation to exome array variants. Instead, we would like to point to our companion protein GWAS paper (MAF > 0.01) reporting a comprehensive colocalization analysis and instead emphasize the exome array study's uniqueness. Also, we have now included two-sample MR analysis on a subset of proteins that have available GWAS summary statistics data and are highlighted in the main text. See our detailed response to the following comment about the two-sample MR analysis.

Comment 5. The authors state that they “highlight how triangulation of data from different sources can link genetics, protein levels and disease(s)”. I would have expected a systematic Mendelian Randomization study here, not just a report of the overlap between pQTLs and GWAS hits.

Response: While rare and low-frequency variants often have significant disease effects, their stated variance (r^2) or F-statistic is relatively low, limiting their use for efficient MR analysis⁴. GWAS-based genome arrays measure mostly common alleles, as mentioned above, and imputations using the 1000G reference panel most effectively capture data on other common rather than rare and low-frequency alleles⁴, which is actually one of the reasons for the prior version of our paper not including colocalization and MR analysis, in addition to sparse genomic coverage of the exome array. However, we believe that the two-sample MR analysis is an important addition to the causal interpretation of the relationship between genotypes, serum proteins and disease and in order to address this we have used available GWAS summary statistics data to include such analysis on selected examples highlighted in the main text.

We applied the “TwoSampleMR” R package⁵ to perform a two-sample MR analysis to test for causal associations between protein and outcome (protein-to-outcome). For different outcomes we used large-scale GWAS associations for LOAD in Europeans⁶, malignant melanoma in European individuals from the UK biobank data (UKB-b-12915)⁷ and T2D in Europeans⁸. Genetic variants (SNPs) associated with serum protein levels at a genome-wide significant threshold ($P < 5 \times 10^{-8}$) identified in the AGES dataset were used as instruments. The inverse variance weighted (IVW) method⁹ was used for the MR analysis, with P-values < 0.05 considered significant. We preferred *cis*-acting pQTLs for genetic instruments but have included *trans*-acting pQTLs in the case of TREM2 where *cis*-acting instruments are scarce. For the examples presented in the main text, the following results were obtained and demonstrated in new figures **Fig. 3d**, **Fig. 4e** and **Fig. 5b**:

1. Using *trans*-acting instruments at the *MS4A4A/MS4A6A* LOAD locus in conjunction with a

cis-acting instrument at the TREM2 locus (**Figure 3d**), variable serum TREM2 levels were found to be causally related to LOAD ($P = 7.6 \times 10^{-5}$), demonstrating that TREM2 directly contributes to the risk of LOAD at these two loci. Also, the MR analysis revealed a significant causal relationship between TREM2 and LOAD using only *trans*-acting instruments at the *MS4A4A/MS4A6A* locus (see below).

Fig. 3d. Scatterplot for the TREM2 protein supported as having a causal effect on LOAD in a two sample MR analysis. The figure demonstrates the estimated effects (with 95% confidence intervals) of their respective *cis*- and *trans*-acting genetic instruments on the serum TREM2 levels in AGES-RS (x-axis) and risk of LOAD through a GWAS by Kunkle et al.⁶ (y-axis) using 21,982 LOAD cases and 41,944 controls. The line indicates the inverse variance weighted causal estimate ($\beta = -0.226$, $SE = 0.057$, $P = 7.6 \times 10^{-5}$).

Response Figure 1. Scatterplot for the TREM2 protein in a two-sample MR analysis. The figure demonstrates the estimated effects (with 95% confidence intervals) of their respective *trans*-acting genetic instruments at chromosome 11, on the serum TREM2 levels in AGES-RS (x-axis) and risk of LOAD *via* a GWAS by Lambert et al.¹⁰ (y-axis) with 17,008 LOAD cases and 37,154. The line indicates the inverse variance weighted causal estimate ($\beta = -0.208$, $SE = 0.036$, $P = 7.2 \times 10^{-9}$).

2. Through a two-sample MR analysis using *cis*-acting instruments from the exome array AGES-RS data and a GWAS of T2D in Europeans⁸ with 74,124 T2D cases and 824,006 controls, we find that the MR analysis is supporting a causal relationship between SVEP1 and T2D (**Fig. 4e**).

Fig. 4e. Scatterplot for the SVEP1 protein supported as having a causal effect on T2D in a two-sample MR analysis. The figure demonstrates the estimated effects (with 95% confidence intervals) of the SNP effect on serum SVEP1 levels and T2D from a GWAS in Europeans⁸ (y-axis) with 74,124 T2D patients and 824,006 controls. The line indicates the inverse variance weighted causal estimate ($\beta = 0.105$, $SE = 0.024$, $P = 1.2 \times 10^{-5}$).

3. Serum levels of ASIP were found to be causally related to malignant melanoma ($P = 4.8 \times 10^{-26}$) using *cis*-acting pQTL instruments in the AGES-RS and a GWAS on malignant melanoma from the UKBB data⁷ of 3,598 melanoma cases and 459,335 controls (**Fig. 5b**).

Fig. 5b. Scatterplot for the ASIP protein supported as having a causal effect on malignant melanoma in a two sample MR analysis. The figure demonstrates the estimated effects (with 95% confidence intervals) of their respective genetic instruments on the serum ASIP levels in AGES (x-axis) and risk of melanoma in GWAS by UK biobank data (UKB-b-12915)⁷ (y-axis) that included 3,598 melanoma cases and 459,335 controls. The line indicates the inverse variance weighted

causal estimate ($\beta = 0.0025$, $SE = 0.0002$, $P = 4.8 \times 10^{-26}$).

The two-sample MR analysis is now described in Methods, and the rationale for using MR analysis, as well as the subsequent positive TREM2, SVEP1, and ASIP results, have been incorporated into the main text:

Methods page 13, lines 313-320:

“We applied the “TwoSampleMR” R package⁵ to perform a two-sample MR analysis to test for causal associations between protein and outcome (protein-to-outcome). For different outcomes we used GWAS associations for LOAD in Europeans⁶, malignant melanoma in European individuals from the UK biobank data (UKB-b-12915)⁷ and T2D in Europeans⁸. Genetic variants (SNPs) associated with serum protein levels at a genome-wide significant threshold ($P < 5 \times 10^{-8}$) identified in the AGES dataset and filtered to only include uncorrelated variants ($r^2 < 0.2$) were used as instruments. The inverse variance weighted (IVW) method⁹ was used for the MR analysis, with P -values < 0.05 considered significant.”

Main text (page 5, lines 125-128):

“Although data triangulation can be used to infer directional consistency, it cannot determine whether the relationship is causal or reactive to a specific outcome. As a result, we used two-sample Mendelian randomization analysis (MR) on highlighted examples to test support for a protein's causality to an outcome.”

Main text (page 6, lines 148-151):

*“Furthermore, a two-sample MR analysis using genetic instruments across the TREM2 and MS4A4A/MS4A6A loci and GWAS associations for LOAD in Europeans as outcome⁶, provided evidence that variable TREM2 protein levels are causally related to LOAD ($P = 7.6 \times 10^{-5}$) (**Fig 3d**)”*

Main text (page 7, lines 168-171):

*“Given the currently available GWAS summary statistics, a two-sample MR analysis using cis-variants on chromosome 9 for SVEP1 as instruments and a GWAS associations for T2D⁸ support a causal relationship of SVEP1 with T2D (**Fig. 4e**), but not with CHD¹¹ or systolic blood pressure¹² ($P > 0.05$).”*

Main text (page 8, lines 186-189):

*“Importantly, we found that serum ASIP levels were supported as causally related to malignant melanoma ($P = 4.8 \times 10^{-26}$) using a two-sample MR analysis on the protein-to-outcome causal sequence of events (**Fig. 5b**)”*

Main text (page 8, lines 197-198):

“Due to a lack of available and powered GWAS summary statistics data, we were unable to formally test the causality of these proteins to colorectal cancer.”

Main text (page 10, lines 235-239):

“Instead, multi-omics data triangulation to infer consistency in directionality, the approach used in the present study, can enhance confidence in the causal call and offer insights and guidelines for experimental follow-up studies. In fact, the causal calls for TREM2 (LOAD), SVEP1 (T2D) and ASIP (melanoma) were validated, using a two-sample MR analysis.”

Comment 6. Epitope-changing variants can affect aptamer binding and lead to pQTL associations that are not reflected in changed protein levels. The authors write “systematic conditional and colocalization analyses in causality testing using the aptamer-based technology have shown that pQTLs driven by common missense variants being artefactual is an unlikely event [refs 11&45]”. I disagree with the generality of this statement. Refs 11&45 acknowledge the existence of epitope effects, but they don’t conclude that they are unlikely events – they are unlikely if the variant colocalizes with a GWAS signal on another trait, which is what the authors do not provide here. In addition, coding variants that can lead to epitope effects are enriched on the Exome array.

Response: We agree with the Reviewer's proposal and have updated the related text (page 9, lines 225-230) accordingly, in line with the previous citations:

“ Protein coding variants may cause technical artifacts in both affinity proteomics and mass spectrometry^{13,14}. Systematic conditional and colocalization studies have shown, however, that pQTLs powered by common missense variants being artifactual are not a common event using the aptamer-based technology^{15,16}, however, given the enrichment of missense variants in the present study, it may occur in some cases“

Comment 7. In summary, while I am generally positive about any new proteomics GWAS, I am a bit hesitant in recommending this paper for publication as it stands. The paper would be much stronger if it integrated the full genotyping data set and then used this as a basis to test for signal colocalization and Mendelian randomization.

Response: We refer to our responses to this comment above, as well as our response to a related comment from Reviewer 3 (see below). Furthermore, wherever GWAS summary statistics data is available, we have performed two-sample MR analyses on the proteins highlighted in the main text. Also, we refer to our GWAS companion paper, which includes extensive colocalization and MR analysis and is being submitted as a separate complementary study, as previously noted.

Reviewer #2

This study examined the overlap and association of genetic disease signatures with circulating serum proteins using the AGES Reykjavik cohort. Overall, this study was well designed, and the manuscript was well-written and interesting. The study makes nice use of data integration to provide supporting evidence for the conclusions that are drawn. The results of the analysis lend support to a growing body of evidence that circulating proteins not only act as biomarkers but in some cases mediate or cause disease. The results are suggestive of the possibility of targeted monitoring of serum protein content for people with certain risk alleles. However, despite the strengths of this article I do have some minor comments.

Response: We appreciate the Reviewer's positive comments on our work.

Comment 1. Lines 100-101: A Fisher's exact test was used, which I acknowledge is fairly common in enrichment tests. However, the independence assumption is not very likely to hold in this case, especially given the later claims of pleiotropy. A permutation or bootstrap test would be likely be a better choice.

Response: The Reviewer is right in pointing out that the precise Fisher test does not adequately address the presumption of data independence. In response to the Reviewer's suggestion, we ran a permutation test to determine if pQTLs are more common in secreted proteins than non-secreted proteins irrespective if they are *cis* or *trans* acting. Here, we wish to test whether the percentage of secreted proteins among pQTLs is equal to the percentage of secreted proteins among non-pQTLs. Our null and alternate hypotheses are thus,

$$H_0: P(\text{pQTL} \mid \text{Secreted}) = P(\text{pQTL} \mid \text{Not Secreted}) \quad H_1: P(\text{pQTL} \mid \text{Secreted}) > P(\text{pQTL} \mid \text{Not Secreted})$$

We performed 10,000 permutations to obtain the empirical distribution of the χ^2 test of equality of proportions. We then compared the test statistics calculated from our data to the quantiles of this distribution to obtain $P(\text{Data} \mid H_0)$. The **Supplementary Fig. S1** below shows the empirical distribution of the test statistic as a histogram and the observed statistics calculated from our data as a vertical line. Of 10,000 permutations none gave a value greater than the observed statistic leading us to $P\text{-value} = P(\text{Data} \mid H_0) < 0.0001$.

Supplementary Fig. S1 now depicts these findings (see also below). Further, following amendments were made to the main text at pages 4-5, lines 105-110 and Methods at page 13, lines 306-312:

Main text pages 4-5, lines 105-110:

*“Secreted proteins were enriched for pQTLs ($P\text{-value} < 0.0001$) as compared to non-secreted proteins (**Fig. 1a**), using 10,000 permutations to obtain the empirical distribution of the χ^2 test of equality of proportions (**Methods** and **Supplementary Fig. S1**), which may indicate that proteins bound for the systemic environment are under greater genetic regulation than other proteins identified by the current platform“*

Methods page 13, lines 306-312:

“To test whether the percentage of secreted proteins among pQTLs is equal to the percentage of secreted proteins among non-pQTLs, 10,000 permutations were performed to obtain the empirical distribution of the χ^2 test of equality of proportions. Our null and alternate hypotheses are,

$$H_0: P(\text{pQTL} | \text{Secreted}) = P(\text{pQTL} | \text{Not Secreted}) \quad H_1: P(\text{pQTL} | \text{Secreted}) > P(\text{pQTL} | \text{Not Secreted})$$

The test statistics calculated from our data was compared to the quantiles of this distribution to obtain $P(\text{Data}|H_0)$ “

Supplementary Fig. S1. Empirical distribution of the test statistic as a histogram and the observed statistics calculated from our data as a vertical line. 10,000 permutations were performed to obtain the empirical distribution of the χ^2 test of equality of proportions of pQTLs among secreted versus non-secreted proteins. Here, the test statistics calculated from our data to the quantiles of this distribution to obtain $P(\text{Data}|H_0)$ were compared. Of 10,000 permutations none gave a value greater than the observed statistic leading us to a P-value = $P(\text{Data}|H_0) <$

0.0001.

Comment 2. Lines 104-106: discuss overlap of pQTLs with known GWAS loci, but the reasoning here seems a bit circular. Isn't the HumanExome BeadChip enriched for known GWAS loci? My point is not that the results are not valid or interesting, simply that this overlap in and of itself may not be.

Response: We are happy to clarify this issue. The exome array is not enriched for known GWAS risk loci: among the 244,883 screened variants the Illumina HumanExome BeadChip is highly enriched for rare and low-frequency variants selected from whole-exome and whole-genome sequencing of 12,000 individuals¹⁷. However, during the development of the platform, 4654 common variants selected from the NHGRI GWAS catalog (accessed 2016) were included to the platform¹⁷, as well as mitochondria and ancestry related markers. As a result, only 1.9% of all variants identified by the exome array are common and known GWAS loci. We should point out that we did not use any statistical analysis to address the high overlap of pQTLs and up-to-date genetic risk factors for a variety of disease-related phenotypes. Thus, we have altered the wording in the main text to avoid any confusion (page 5, lines 112-115):

“Next, we cross-referenced all the 5472 genome-wide significant pQTLs with a comprehensive collection of genetic loci associated with diseases and clinical traits from the curated PhenoScanner database¹⁸, revealing that 60% of all pQTLs were linked to at least one disease-related trait (**Supplementary Tables S4**)“

To tone this down, we have also modified the title of our article as follows: “*Coding and regulatory variants affect serum protein levels and common disease*“

Comment 3. Lines 111-112: suggest “that greater regulatory pleiotropy of pQTLs is associated with greater chance of disease trait pleiotropy”. However, the phenotypes from PhenoScanner are not all independent (e.g. some are merely different ways of diagnosing the same condition) and I’m not sure about the independence of the serum proteins either. This makes correlation a tricky measure and it is not clear to me if this was taken into account. The Spearman correlation of 0.22 is not necessarily so high as to provide a lot of buffer on this front.

Response: We agree with the opinion of the Reviewer that resolving this issue with the current data is not straightforward. In addition, provided that the entire association analysis has been revisited as detailed in our responses to Reviewers 1 and 3, and with the additional two-sample MR analysis, we conclude that this is less relevant. We would like to point out that we have another paper entitled “*A genome-wide association study of serum proteins reveals shared loci with common diseases*” submitted to *Nature Communications* which addresses potential pleiotropy of the serum proteome in a much more comprehensive manner. Accordingly, the text referring to the global pleiotropy test with respect to serum proteins and diseases has been removed. Accordingly we have removed former **Fig. 2a**. However, we retained the example of the influence of a variant rs2251219 on many proteins and related to many diseases of various etiologies, and instead emphasized our previous observation that genetic loci affecting many serum proteins show pleiotropy in relation to disease¹. Because the entire study was revisited, **Fig. 2** had to be re-created because the association with NPPB did not remain study-wide significant. In other words, rs2251219 now affects 13 proteins rather than 14 as previously. The text has, as follows, been updated (page 5, lines 115-119):

“We have shown in our previous studies that genetic loci affecting several serum proteins exhibit pleiotropy in relation to complex diseases¹. An example of a possible pleiotropic effect mediated by the variant rs2251219 within the gene PBRM1 affecting multiple proteins and sharing genetics with various diseases and clinical features is illustrated in Fig. 2.”

A new **Supplementary Fig. S2** (below) has also been generated to highlight the relationship between all of the proteins and some quantitative clinical outcomes associated with rs2251219.

Supplementary Fig. S2. A Spearman rank correlation between all proteins as well as some quantitative traits including body mass index (BMI, kg/m²), visceral adipose tissue (VAT, measured *via* computed tomography) and hematocrit (HCT), that were associated with rs2251219.

Comment 4. Lines 182-183: states “60% of the serum proteome that is under genetic controls shares genetics with reported clinical traits”. However, the entire serum proteome is not measured, and it is not clear what proportion is under genetic control from these results. I would just reword this a bit.

Response: We agree with the Reviewer and refer to our previous answers and an updated version of the text below in this respect (page 9 lines 218-219):

“We report here that many of the measured serum proteins under genetic control share genetics with a variety of clinical features, including major diseases arising from various body tissues”

Reviewer #3

In their manuscript titled "Human serum proteome profoundly overlaps with genetic signatures of disease", Emilsson et al. studied associations of 55932 low-frequency and common exonic variants with 4782 protein measures in serum samples of 5457 individuals from AGES Reykjavik study. The authors identified 5553 variants associating with levels of 1931 serum proteins, and they further characterized the overlap of the genetic associations between proteins and human diseases.

The manuscript has multiple positive aspects. It provides a larger sample size and a larger number of proteins studied than the previous reports describing genome-wide associations of human blood proteome^{16,19}, it is well written, and it would be of interest to researchers in multiple fields of science.

Response: We thank the Reviewer for his/her positive comments on our work

Unfortunately, the reviewer is not convinced if the manuscript fulfills the novelty criteria that can be expected from a high-quality journal like Nature Communications, as the data has been published as part of the previous work by the authors¹. Although the focus of the previous paper is in describing serum protein networks and the description of the genetic associations is somewhat in the background, the same proteome and exome data in 5457 participants of AGES have been analyzed. The high overlap between the two papers is highlighted by the fact that the text in the Methods paragraph is partially identical to the text in Supplemental Methods of the previous publication¹, (lines 264-269, for example).

Response: We respectfully disagree with the statement of the Reviewer that in our Science article¹ the exome data in 5457 AGES participants was evaluated. That is incorrect because it was not available at the time. In the previous study¹, we used a genotyping platform (Illumina 370CNV) that measures common ($MAF \geq 0.05$) variants in 3219 individuals, while the exome array is enriched for rare and low-frequency variants and was evaluated in all 5457 AGES individuals with serum protein measures. In addition, when we submitted our paper to *Nature Communications*, we had only published a single paper explaining the proteomics platform, so we assumed that many readers would benefit from a brief summary in the Method section. That description, however, is nowhere close to the details given in Emilsson et al.¹ To avoid repetitive explanations of the platform, however, we have rewritten this Method section as follows:

Methods page 11-12, lines 272-283

"Each protein has its own detection reagent selected from chemically modified DNA libraries, referred to as Slow Off-rate Modified Aptamers (SOMAmers)²⁰. The design and quality control of the SOMApanel platform's custom version to include proteins known or predicted to be present in the extracellular milieu have been described in detail elsewhere¹. Briefly though, the aptamer-based platform measures 5034 protein analytes in a single serum sample, of which 4782 SOMAmers bind specifically to 4137 human proteins (some proteins are identified by more than one aptamer) and 250 SOMAmers that recognize non-human targets (47 non-human vertebrate proteins and 203 targeting human pathogens)¹. Consistent target specificity across the platform was indicated by direct (through mass spectrometry) and/or indirect validation of

the SOMAmers¹. Both sample selection and sample processing for protein measurements were randomized, and all samples were run as a single set to prevent batch or time of processing biases“

In terms of the overlap between the two papers, we refer to our more detailed reply to Reviewer 1 above. We also refer to all of the new analyses that have been included in the most recent edition and have contributed to the current report's novelty and greater impact.

Also, the following issues should be addressed:

Comment 1. The association signals should be better characterized to identify the number of independent associations within a genomic region. For example, in lines 147-148, the authors describe that they “found eight different missense mutations in SVEP1” – to determine if these missense variants associate with SVEP1 level independently from each other, conditional analyses should be performed. One option to do this is with GCTA software using summary-level data and linkage disequilibrium (LD) matrix^{21,22}. Additionally, variant annotation tools, such as SIFT, PolyPhen2, or MutationTaster (all implemented in ANNOVAR²³), could be used to gain insights about the consequences of the variants associated with protein levels and to rank adjacent variants to identify the potentially causal ones.

Response: We would like to thank the Reviewer for his/her helpful suggestions. As detailed elsewhere in our responses, the full analysis of the link between the exome array and the serum protein levels has been revised to take into account the potential effect of the population structure, and a conditional analysis has been carried out to define independent association signals. For the conditional analysis we used the GCTA-COJO software^{21,22} as suggested, and the new results are shown in a new **Supplementary Table S2**. For example, instead of 15 exome array variants associated with SVEP1 at $P < 1 \times 10^{-6}$ (**Supplementary Table S1**), five independent variants were linked to SVEP1 using a conditional analysis at $P < 1 \times 10^{-6}$ (see **Supplementary Table S2**). Overall, compared to the 10,200 exome array variants associated with 3107 aptamers (2780 proteins with unique gene symbols) at $P < 1 \times 10^{-6}$ shown in **Supplementary Table S1** prior to the conditional analysis, there are 5259 independent associations signals for the same set of protein targets (**Supplementary Table S2**). Using the study-wide significant threshold $P < 1.92 \times 10^{-10}$, 5472 exome array variants were associated with 2135 protein targets prior to the conditional analysis, of which 2019 were independent association signals for the same set of proteins.

Conditional analysis is described in the section Method page 12 lines 295-302:

*“Independent genetic signals were found through a stepwise conditional and joint association analysis for each protein analyte separately with the GCTA-COJO software^{21,22}. We conditioned on the current lead variant listed in **Supplementary Table S1**, defined as the variant with the lowest P-value, and then kept track of any new variants that were not in LD (the default GCTA-COJO option $r^2 < 0.9$ for colinearity) with previously chosen lead variants and reported findings at P-value $< 1 \times 10^{-6}$ (**Supplementary Table S2**). In the joint model all conditionally significant SNPs for each protein analyte were combined in the regression model.”*

As regards the pathogenicity of the exome array variants, a new **Supplementary Table S3** was produced to assess pathogenicity of all independent variants of the conditional analysis listed in

Supplementary Table S2. Here we applied the software suite Ensembl Variant Effect Predictor (VEP)^{24,25} via the variant annotation integrator (<http://genome.ucsc.edu>), that has been widely used for pathogenicity annotation and prediction, for example for genomic variants resulting from DNA deep sequencing efforts²⁶. The VEP uses different pathogenic predictors and compares well with tools such as ANNOVAR²³, but contains more features (see method comparison in **Table 1** in McLaren et al.²⁴), and is, unlike ANNOVAR, free of registration. We note that the different pathogenic predictors do not always agree, thus making it difficult to generate a single rank score based on prediction of pathogenicity alone. Instead, VEP offers different prediction scores including the Likelihood Ratio Test (LRT)²⁷, Variant Effect Scoring Tool (VEST)²⁸, MutationAssessor²⁹ and MutationTaster³⁰ (see **Supplementary Table S3**).

The use of the VEP tool for pathogenic annotation of variants in **Supplementary Table S3** is described in main text on page 5, lines 110-112:

“Supplementary Table S3 summarizes various pathogenicity prediction scores for all study-wide significant pQTLs in Supplementary Table S2, using the Ensembl Variant Effect Predictor (VEP)^{24,25}”

And in Methods page 13, lines 303-306

“Supplementary Table S3 summarizes, through use of VEP^{24,25}, various pathogenicity prediction scores for all study-wide significant pQTLs in Supplementary Table S2, including the Likelihood Ratio Test (LRT)²⁷, Variant Effect Scoring Tool (VEST)²⁸, MutationAssessor²⁹ and MutationTaster³⁰”

Comment 2. The results presented in the manuscript are based on a single population. According to the reporting summary, independent replication was not possible due to a lack of suitable data in other cohorts. The authors should investigate if the loci reported here were reported in the previous studies (for the overlapping proteins)^{16,19}. Replication of the known genetic associations provides assurance also for the novel associations.

Response: In our previous study¹, for those proteins that were examined in both the external and internal study populations, we tested a replication of our findings related to the genetics of serum protein levels. This included replication through the same and different proteomics platforms. In summary, we find that on average, 80% of *cis* effects and 74% of *trans* effects replicate (see for instance **Supplementary Table S18** in Emilsson et al.¹). The Reviewer is correct in arguing that such a form of validation offers assurance for novel findings. As regards comparing findings using the exome array platform which is enriched for rare and low-frequency variants with the other platforms including only common variants at $MAF \geq 0.05$ the comparison is not straightforward. We refer to our reply to Reviewer 1's comment 2 above, highlighting that only 12.0% to 22.8%, depending on P-value threshold, of the genetic associations found in the present study, were reported in our previous work¹. We included data from other proteomics studies instead of just comparing new findings to Sun et al.¹⁶ and Suhre et al.¹⁹. Using the independent study-wide significant pQTLs in **Supplementary Table S2** and LD of $r^2 < 0.5$, we find that the current study generates 76.0% new SNP-to-protein associations compared to Sun et al.¹⁶ and 60.1% percent compared to the majority (any) of studies published to date including that of Suhre et al.¹⁹ (see **Response Fig. 2**). Using a more conservative assessment of novelty as per locus rather than a specific LD threshold between variants and restricting to SNPs that are more

than 500kb physically distant from the reported lead SNP for the same protein, 1670 SNP-to-protein associations are novel in the current study.

Response Fig. 2. Using the independent study-wide significant pQTLs in **Supplementary Table S2**, we find that the current study produced 76.0% novel SNP-to-protein associations compared to that published in Sun et al.¹⁶ and 60.1% when compared to majority of studies (including Suhre et al.¹⁹) published to date (i.e. any).

We highlight this in the main text on page 9, lines 209-217 as follows:

“Previously, we discovered that 80% of cis pQTL effects and 74% of trans pQTL effects were replicated across populations and proteomics platforms measuring common variants. Given that the exome array platform is enriched for rare and low-frequency variants, a comparable test of replication is not straightforward. Examining the proteins and variants measured across studies, we find that 76.0% of SNP-to-protein associations are novel in the present study when compared to, say, Sun et al.¹⁶, and 60.1% are novel when compared to majority of studies published to date (Supplementary Table S5), for all independent associations in the current study and LD of $r^2 < 0.5$ between study specific markers.”

In Methods, page 14, lines 328-342:

“We compared our pQTL results to 19 previously published proteogenomic studies (Supplementary Table 5), including the protein GWAS in the INTERVAL study¹⁶, and our previously reported genetic analysis of 3,219 AGES cohort participants¹. In the previous proteogenomic analysis of AGES participants, one cis variant was reported per protein using a locus-wide significance threshold, as well as cis-to-trans variants at a Bonferroni corrected significance threshold. Due to these differences in reporting criteria, we only considered the associations in previous AGES results that met the current study-wide P-value threshold. For all other studies we retained the pQTLs at the reported significance threshold. In addition, we performed a lookup of all independent pQTLs from the current study available in summary statistics from the INTERVAL study, considering them known if they reached a study-wide significance in their data. We calculated the LD structure between the reported significant variants for all studies, using 1000 Genomes v3 EUR samples, but using AGES data when comparing to previously reported AGES results. We considered variants in LD at $r^2 > 0.5$ to

represent the same signal across studies. Comparison was performed on protein level, by matching the reported Entrez gene symbol from each study.”

Comment 3. The authors mention the “triangulation of data” on a few occasions. Still, they have not performed Mendelian randomization analyses, which would help to combine information from multiple association tests to causal estimates between protein levels and human diseases^{5,16}. It is probably beyond the scope of this manuscript to perform Mendelian randomization in all significant loci, but it would be beneficial to report causal estimates for the examples highlighted in the manuscript.

Response: In our reply to Reviewer 1 above we noted that low frequency variants are not well represented in public domain GWAS summary statistics data. Nonetheless, we highlight a successful two-sample MR analysis examining the causal relationship between the protein and outcome (protein-to-outcome) for the examples highlighted in the main text, TREM2 (LOAD), SVEP1 (T2D), and ASIP (malignant melanoma).

Comment 4. Population stratification is a major confounder in genetic association studies³¹, and it can affect the validity of the association results even in populations considered to be relatively homogenous, such as the Icelanders³². The authors have not corrected the genetic association tests for population stratification.

Response: We would like to thank the Reviewer for making a valid point here. The reviewer is right in suggesting that the AGES-RS cohort is drawn from a relatively homogeneous population, and we point out that each participant is of Northern European descent. Because we can not rule out possible confounding of substructure effects on the outcome, i.e. serum protein levels, we have revisited the entire association study obtaining residuals after adjustment for sex, age, and potential population stratification using principal component (PCs) analysis³¹. Therefore, for all single-variant associations to serum protein levels tested under the additive genetic model we adjusted for five genetic PCs as follows: protein ~ SNP + age + sex + PC1 + PC2 + ...PC5. The first five PCs were chosen based on the criterion that after the fifth PC, the proportional variance explained by the PCs begins to flatten out. In the current version, we report variants to protein associations at $P < 1 \times 10^{-6}$ for suggestive evidence and Bonferroni correction for multiple comparisons by adjusting for the 54,469 variants and 4782 human protein analytes where single variant associations with $P < 1.92 \times 10^{-10}$ were considered study-wide significant (**Supplementary Table S1**).

Minor corrections/comments:

Minor Comment 1. The authors report the number of exome array variants (5553) associating with levels of serum proteins. It would be helpful to describe further the number of independent genomic regions associating with blood protein levels.

Response: We now provide conditional and joint association analysis, as evident in our reaction to comment 1 by the same Reviewer, and refer to that reply for more information.

Minor Comment 2. Technically, “rs123456” is not a locus - the rs number is a unique identifier for a sequence variant in a locus (for example, rs2251219 is an identifier for allelic variation T/G/C in locus chr3:52550771). Usually, in genetic studies, “locus” refers to a genomic region larger than a single nucleotide. Please edit the incorrect expressions, such as “the locus rs2251219” on line 114.

Response: We thank the Reviewer for pointing out this error and have, wherever possible, changed the wording accordingly (corrections made in nine places of main text and figure legends including rs2251219).

Minor Comment 3. The methods section does not describe the software used for statistical analyses.

Response: a description of the software used for the statistical analysis has been included in the Method section (page 13, line 326 and 327):

“All statistical analysis was performed using R version 3.6.0 (R Foundation for Statistical Computing, Vienna, Austria)”

Minor Comment 4. In case the text above Figure 1b indicates gene names, the font should be italic.

Response: Given the entire study was revisited, a new and slightly different **Fig. 1b** is now included, with the font of genes highlighted in italic.

Minor Comment 5. In Figures 3a and 4a, instead of showing Manhattan plots, it would be more useful to show the regional association plots of the significant association signals. These plots should include information about the LD structure. With the exome-wide data, the number of SNPs in the regional plots will be lower than in the case of genome-wide data, but they can still be very informative^{33,34}.

Response: Manhattan plots were presented in the main text and regional plots in the supplementary data in both papers cited by the Reviewer. We would like to retain in the main text the Manhattan plots and demonstrate the regional plots in the new **Supplementary Material** as **Supplementary Figures 3 and 4** (see below). For the generation of the regional plots we used the LocusZoom³⁵.

a

b

Supplementary Fig. S3. TREM2 regional plots (LocusZoom) based on exome array variants at chromosomes 6 and 11.

a**b**
Supplementary Fig. S4. SVEP1 regional plots (LocusZoom) based on exome array variants at chromosomes 1 and 9.

Minor Comment 6. In Figure 3d, it would be helpful to show the correlation coefficients on the plot.

Response: The correlation coefficients have been added to the previous **Figure 3d** (which has been replaced by the MR results) and are now referred to as **Supplementary Figure 4**, as shown below.

Supplementary Fig. S4. The graph shows the Spearman rank correlation between the four serum proteins affected by the two LOAD risk variants, rs75932628 and rs610932. The correlation matrix's upper triangle depicts the beta-values, while the lower triangle highlights the P-values.

References

- 1 Emilsson, V. *et al.* Co-regulatory networks of human serum proteins link genetics to disease. *Science* **361**, 769-773, doi:10.1126/science.aaq1327 (2018).
- 2 Lamb, J. R., Jennings, L. L., Gudmundsdottir, V., Gudnason, V. & Emilsson, V. It's in Our Blood: A Glimpse of Personalized Medicine. *Trends Mol Med*, doi:10.1016/j.molmed.2020.09.003 (2020).
- 3 Huang, J. *et al.* Improved imputation of low-frequency and rare variants using the UK10K haplotype reference panel. *Nat Commun* **6**, 8111, doi:10.1038/ncomms9111 (2015).
- 4 Swerdlow, D. I. *et al.* Selecting instruments for Mendelian randomization in the wake of genome-wide association studies. *Int J Epidemiol* **45**, 1600-1616, doi:10.1093/ije/dyw088 (2016).
- 5 Hemani, G. *et al.* The MR-Base platform supports systematic causal inference across the human phenome. *Elife* **7**, doi:10.7554/eLife.34408 (2018).
- 6 Kunkle, B. W. *et al.* Genetic meta-analysis of diagnosed Alzheimer's disease identifies new risk loci and implicates A β , tau, immunity and lipid processing. *Nat Genet* **51**, 414-430, doi:10.1038/s41588-019-0358-2 (2019).
- 7 Sudlow, C. *et al.* UK biobank: an open access resource for identifying the causes of a wide range of complex diseases of middle and old age. *PLoS Med* **12**, e1001779, doi:10.1371/journal.pmed.1001779 (2015).
- 8 Mahajan, A. *et al.* Fine-mapping type 2 diabetes loci to single-variant resolution using high-density imputation and islet-specific epigenome maps. *Nat Genet* **50**, 1505-1513, doi:10.1038/s41588-018-0241-6 (2018).
- 9 Burgess, S., Butterworth, A. & Thompson, S. G. Mendelian randomization analysis with multiple genetic variants using summarized data. *Genet Epidemiol* **37**, 658-665, doi:10.1002/gepi.21758 (2013).
- 10 Lambert, J. C. *et al.* Meta-analysis of 74,046 individuals identifies 11 new susceptibility loci for Alzheimer's disease. *Nature Genetics* **45**, 1452-1458, doi:10.1038/ng.2802 (2013).
- 11 Nikpay, M. *et al.* A comprehensive 1,000 Genomes-based genome-wide association meta-analysis of coronary artery disease. *Nat Genet* **47**, 1121-1130, doi:10.1038/ng.3396 (2015).
- 12 Evangelou, E. *et al.* Genetic analysis of over 1 million people identifies 535 new loci associated with blood pressure traits. *Nat Genet* **50**, 1412-1425, doi:10.1038/s41588-018-0205-x (2018).
- 13 Solomon, T. *et al.* Identification of Common and Rare Genetic Variation Associated With Plasma Protein Levels Using Whole-Exome Sequencing and Mass Spectrometry. *Circ Genom Precis Med* **11**, e002170, doi:10.1161/circgen.118.002170 (2018).
- 14 Smith, J. G. & Gerszten, R. E. Emerging Affinity-Based Proteomic Technologies for Large-Scale Plasma Profiling in Cardiovascular Disease. *Circulation* **135**, 1651-1664, doi:10.1161/circulationaha.116.025446 (2017).
- 15 Zheng, J. *et al.* Phenome-wide Mendelian randomization mapping the influence of the plasma proteome on complex diseases. *bioRxiv*, 627398, doi:10.1101/627398 (2019).
- 16 Sun, B. B. *et al.* Genomic atlas of the human plasma proteome. *Nature* **558**, 73-79, doi:10.1038/s41586-018-0175-2 (2018).
- 17 Liu, D. J. *et al.* Exome-wide association study of plasma lipids in >300,000 individuals. *Nature Genetics* **49**, 1758-1766, doi:10.1038/ng.3977 (2017).
- 18 Staley, J. R. *et al.* PhenoScanner: a database of human genotype-phenotype associations. *Bioinformatics* **32**, 3207-3209, doi:10.1093/bioinformatics/btw373 (2016).
- 19 Suhre, K. *et al.* Connecting genetic risk to disease end points through the human blood plasma proteome. *Nature Communications* **8**, 14357, doi:10.1038/ncomms14357 (2017).
- 20 Candia, J. *et al.* Assessment of Variability in the SOMAscan Assay. *Sci Rep* **7**, 14248, doi:10.1038/s41598-017-14755-5 (2017).

- 21 Yang, J. *et al.* Conditional and joint multiple-SNP analysis of GWAS summary statistics identifies additional variants influencing complex traits. *Nat Genet* **44**, 369-375, s361-363, doi:10.1038/ng.2213 (2012).
- 22 Yang, J., Lee, S. H., Goddard, M. E. & Visscher, P. M. GCTA: a tool for genome-wide complex trait analysis. *Am J Hum Genet* **88**, 76-82, doi:10.1016/j.ajhg.2010.11.011 (2011).
- 23 Wang, K., Li, M. & Hakonarson, H. ANNOVAR: functional annotation of genetic variants from high-throughput sequencing data. *Nucleic Acids Res* **38**, e164, doi:10.1093/nar/gkq603 (2010).
- 24 McLaren, W. *et al.* The Ensembl Variant Effect Predictor. *Genome Biol* **17**, 122, doi:10.1186/s13059-016-0974-4 (2016).
- 25 McLaren, W. *et al.* Deriving the consequences of genomic variants with the Ensembl API and SNP Effect Predictor. *Bioinformatics* **26**, 2069-2070, doi:10.1093/bioinformatics/btq330 (2010).
- 26 Wright, C. F. *et al.* Genetic diagnosis of developmental disorders in the DDD study: a scalable analysis of genome-wide research data. *Lancet* **385**, 1305-1314, doi:10.1016/s0140-6736(14)61705-0 (2015).
- 27 Chun, S. & Fay, J. C. Identification of deleterious mutations within three human genomes. *Genome Res* **19**, 1553-1561, doi:10.1101/gr.092619.109 (2009).
- 28 Carter, H., Douville, C., Stenson, P. D., Cooper, D. N. & Karchin, R. Identifying Mendelian disease genes with the variant effect scoring tool. *BMC Genomics* **14 Suppl 3**, S3, doi:10.1186/1471-2164-14-s3-s3 (2013).
- 29 Reva, B., Antipin, Y. & Sander, C. Predicting the functional impact of protein mutations: application to cancer genomics. *Nucleic Acids Res* **39**, e118, doi:10.1093/nar/gkr407 (2011).
- 30 Schwarz, J. M., Rödelberger, C., Schuelke, M. & Seelow, D. MutationTaster evaluates disease-causing potential of sequence alterations. *Nat Methods* **7**, 575-576, doi:10.1038/nmeth0810-575 (2010).
- 31 Price, A. L. *et al.* Principal components analysis corrects for stratification in genome-wide association studies. *Nat Genet* **38**, 904-909, doi:10.1038/ng1847 (2006).
- 32 Helgason, A., Yngvadóttir, B., Hrafnkelsson, B., Gulcher, J. & Stefánsson, K. An Icelandic example of the impact of population structure on association studies. *Nat Genet* **37**, 90-95, doi:10.1038/ng1492 (2005).
- 33 Chang, J. *et al.* Exome-wide analyses identify low-frequency variant in CYP26B1 and additional coding variants associated with esophageal squamous cell carcinoma. *Nat Genet* **50**, 338-343, doi:10.1038/s41588-018-0045-8 (2018).
- 34 Natarajan, P. *et al.* Multiethnic Exome-Wide Association Study of Subclinical Atherosclerosis. *Circ Cardiovasc Genet* **9**, 511-520, doi:10.1161/circgenetics.116.001572 (2016).
- 35 Boughton, A. P. *et al.* LocusZoom.js: Interactive and embeddable visualization of genetic association study results. *Bioinformatics*, doi:10.1093/bioinformatics/btab186 (2021).

Reviewers' Comments:

Reviewer #1:

Remarks to the Author:

The authors have responded to my previous points.

Reviewer #2:

Remarks to the Author:

The authors have done a good job of responding to all reviewer comments and I have no further concerns.

Reviewer #3:

Remarks to the Author:

Please see my comments in the attached files.

I am delighted to see that the authors have fixed the analytical issue and incorporated genetic PCs in their models. It is also positive to see that they have added causal evidence as suggested by the Reviewer #1 and myself. However, as much as from the methodical point of view the manuscript seems now mostly competent, I am not convinced if it fulfils the high standards for publication in Nature Communications due to multiple inaccuracies recognized during re-review.

I still have major concerns regarding the novelty of this work, and the announcement of yet another association study in the very same population does not add to my excitement. Firstly, in general, high-quality genetic association studies should have evidence from more than one population – the unique proteome data is the only asset of this study that single population setting can be somewhat excused, but I am not convinced that it is valuable enough for three high-impact publications (2018 Science, exome manuscript, GWAS manuscript). Furthermore, I feel that the authors have not adequately addressed the issues related to the overlap between the genotyping chips used in the present manuscript vs. in the 2018 Science paper (the 2nd point by Reviewer #1), and the novelty of the present findings compared with previous reports (the 2nd major point in my original comments). It is obvious that, by analyzing different sets of SNPs, the authors will find different sets of significant SNP-to-protein associations, but the loci may still be the same – hence, professionally conducted GWAS studies typically report novelty on a locus level and not on a variant level. As the sample is the same, and the proteome data is the same, the authors may find some novelty in the genomic regions that are not covered by the genotyping chip used in the 2018 Science paper. The claim that the comparison of the loci is complicated as they analyze rare and low-frequency variants is totally irrelevant and gives me the impression that the authors may lack expertise in genetic studies. Only at the very end of their response, the authors describe that “restricting to SNPs that are more than 500kb physically distant from the reported lead SNP for the same protein, 1670 SNP-to-protein associations are novel”; this is close to what I was looking for; however, instead of reporting novel SNP-to-protein associations, it would be more relevant to get an idea of the number of novel loci, and it is unclear if this comparison also included their previous Science paper or just the papers by Sun *et al.* [1] and Suhre *et al.* [2]. Also, 500kb distance is rather short (sometimes larger distances, such as 1Mb, are used [3]).

To quickly compare the protein-associated loci in the three studies by the authors, I extracted data from the supplemental files of the 2018 Science paper and the two manuscripts submitted to Nature Communications, and produced the attached Manhattan-type plots where each chromosome is plotted separately to allow more clear separation of the key loci; in a typical Manhattan plot, the data points were too densely packed. For each variant, only the most significant association is plotted in order to avoid overlap from the variants showing association with multiple protein. Also, for equality, only associations with p-values below the significance threshold used in the GWAS manuscript ($p < 1.046e-11$) are plotted instead using the varying study-specific significance thresholds. The plots demonstrate that, even if only a small proportion of the variants was analyzed in all the studies, the association peaks locate on the same genomic regions. To define ‘true novel signals’ in their study, the authors should identify the protein-associated loci in the present study, in the recently submitted GWAS, in the previous Science paper, and in other previous proteome-GWASs, and carefully address if the loci they find are overlapping or nonoverlapping with the previously reported loci: reproduction of high-impact papers with essentially same findings should not be allowed. It seems that most of the novelty in their manuscripts is in the downstream analyses (i.e., enrichment, colocalization, Mendelian randomization) and not in the genetic associations – the manuscripts should be clearly structured to avoid this false impression (and most likely only one manuscript would be sufficient to report the results of downstream analyses).

In addition, the following issues gave me the impression that the authors have not finalized the manuscript in a manner that would be appropriate to a high-impact journal:

- In my original Minor Comment 5, I asked to include LD information in the regional association plots. Public data is not ideal for this purpose, especially for rare/low-frequency variants, which can be seen as missing LD information in all the regional association plots. Preferably, the authors should calculate LD based on their genetic data and plot the regional associations using software that allows implementation of your own LD information. Also, the inclusion of

rs-IDs of the key variants in the regional association plots would aid the reader to follow the discussion of the findings; for example, now the reader can only assume that rs75932628 and rs610932 are the lead variants in the TREM2-associated loci, as this is not shown in the figures and not clearly indicated in the text.

- It is impossible to read VEP results in Supplementary Table 3. The authors should see more efforts to ease the reader.
- I am not sure why rs704 (chr17) that associates ($p \sim 1e-12$) with TREM2 level (Supplementary Tables 1 and 2) is not plotted in Figure 3a and is not mentioned in the text with the TREM2 associations in chromosomes 6 and 11.
- Figure legends should describe the methodology used for producing the figure so that the figures are self-explanatory. In this manuscript, figure legends tend to describe results. Key information is missing, including the y-axis units in Figures 3b, 4b, 4c, 5a, and 5d. Are the boxplots shown for adjusted protein levels?
- Supplementary Figure 4: The legend says “The correlation matrix’s upper triangle depicts the beta-values”, but the color map refers to ‘Spearman correlation’ for which the coefficient is called ρ (rho), not beta.
- The authors should go through the content in the Supplemental Tables and check if all column headers are explained sufficiently (for example, I would assume that columns ‘Gene’ and ‘EntrezGeneSymbol’ may cause confusion) and if all the columns need to be repeated in all the tables or if overlapping information could be reduced.
- Lines 159-161: “Overall, we found eight different missense mutations in *SVEP1* that were associated with SVEP1 serum levels (Supplementary Table 1).” Gene name and mutation type are not given in Supplementary Table 1. Is there information missing from this table, or is the reference to the wrong table?
- Reporting of Mendelian randomization results falls somewhat short of the state-of-art. A thorough report would provide the results also in a tabulated form (in the supplement), p-values for horizontal pleiotropy and heterogeneity, and also plots for leave-one-out sensitivity analyses. In the description of the methods, the authors should describe the reference population for calculating LD. Further, by running bidirectional Mendelian randomization, the authors would have an opportunity to distinguish disease biomarkers from causative agents.

Further, the following points were not clear to me:

- Lines 108-110: “This suggests that proteins bound for the systemic environment are subject to more genetic regulation than other proteins identified by the current platform.” – Do you mean to say that the circulating level/quantity of non-secreted proteins is under stricter genetic control than the quantity of secreted proteins? If so, it is problematic that ‘quantity’ is not mentioned, as surely all endogenous proteins are under genetic regulation. Also, it is confusing that the terminology is different from the preceding sentence (does “proteins bound for the systemic environment” refer to non-secreted proteins?).
- Figure 4: Why is the scatter plot presented only for T2D but not CHD and systolic BP? Those could be in the supplement if not in the manuscript main body.
- Lines 130-132 and lines 157-159; I am not sure if I follow the justification for the selection of the key variants presented in the examples. For example, in case of TREM2 association in chr11, the authors highlight an intergenic variant rs610932 ($p_{\text{TREM2.5635-66}}=1e-106$; $p_{\text{TREM2.11851-21}}=3e-46$) instead of rs7232, a missense variant with deleterious prediction ($p_{\text{TREM2.5635-66}}=8e-131$; $p_{\text{TREM2.11851-21}}=2e-54$) that is also included as an instrument in Mendelian randomization. I understand the prior link with LOAD, but isn’t this exactly where the authors would have an opportunity to highlight the advantage of their exonic data? Perhaps they could even suggest something along the lines that “the previously reported association with LOAD in this locus could arise due to coding variants in *MS4A6A*, such as rs7232, a missense variant with deleterious prediction in LD ($r^2 \sim 0.7$) with the LOAD lead variant rs610932” as, based on Supplementary Table S2, the association of rs610932 does not remain significant in the conditional analyses?

References

1. Sun BB, Maranville JC, Peters JE, Stacey D, Staley JR, Blackshaw J, et al. Genomic atlas of the human plasma proteome. *Nature*. 2018;558:73–79.
2. Suhre K, Arnold M, Bhagwat AM, Cotton RJ, Engelke R, Raffler J, et al. Connecting genetic risk to disease end points through the human blood plasma proteome. *Nat Commun*. 2017;8:14357.
3. Liu DJ, Peloso GM, Yu H, Butterworth AS, Wang X, Mahajan A, et al. Exome-wide association study of plasma lipids in >300,000 individuals. *Nat Genet*. 2017;49:1758–1766.

● GWAS ● EXOME ● Reported in both

● GWAS ● EXOME ● Reported in both

● GWAS ● EXOME ● Reported in both

● GWAS ● EXOME ● Reported in both

● GWAS ● Science ● Reported in both

● GWAS ● Science ● Reported in both

● GWAS ● Science ● Reported in both

● GWAS ● Science ● Reported in both

Response to Reviewers

We are pleased to submit our revised manuscript now entitled “*Coding and regulatory variants control serum protein levels and disease*” (NCOMMS-20-16689A) for consideration to be published in *Nature Communications*. We are grateful to reviewers 1 and 2 for positive responses to our previous revision. We carefully considered and responded to each of reviewer 3's additional comments below, as well as improving the clarity of the results (and correcting one mistake). Also addressed is the issue of novelty of this study versus our own GWAS study (NCOMMS-21-24739-T) and previous publications using different comparisons that cover the complexity and depth of information that emerges from the genetics of protein expression studies. As described below, this study provides substantial new information regardless of how this is addressed. We hope you agree that these changes have resulted in a significantly stronger paper.

Please note that we have included a new coauthor Jonmundsson T. who has contributed significantly to the additional analyses and drafting of the revised manuscript. We have added his affiliation to the revised manuscript.

In addition, we'd like to change the title of our paper to "*Coding and regulatory variants control serum protein levels and disease*" rather than "*Coding and regulatory variants affect serum protein levels and common disease*."

Responses below are provided in blue font. Text added to the revised manuscript has been italicized. Page and paragraph numbers listed below refer to the position of the new or modified text in the *clean* version of the revised manuscript (submitted along with a manuscript text file highlighting all changes using the track changes mode).

Reviewers' comments:

Reviewer #1

The authors have responded to my previous points.

Response: We thank the reviewer for positive response to our revision

Reviewer #2

The authors have done a good job of responding to all reviewer comments and I have no further concerns.

Response: We thank the reviewer for positive response to our revision of the manuscript

Reviewer #3

I am delighted to see that the authors have fixed the analytical issue and incorporated genetic PCs in their models. It is also positive to see that they have added causal evidence as suggested by the Reviewer #1 and myself. However, as much as from the methodical point of view the manuscript seems now mostly competent, I am not convinced if it fulfils the high standards for publication in Nature Communications due to multiple inaccuracies recognized during re-review.

Comment 1. I still have major concerns regarding the novelty of this work, and the announcement of yet another association study in the very same population does not add to my excitement. Firstly, in general, high-quality genetic association studies should have evidence from more than one population – the unique proteome data is the only asset of this study that single population setting can be somewhat excused, but I am not convinced that it is valuable enough for three high-impact publications (2018 Science, exome manuscript, GWAS manuscript). Furthermore, I feel that the authors have not adequately addressed the issues related to the overlap between the genotyping chips used in the present manuscript vs. in the 2018 Science paper (the 2nd point by Reviewer #1), and the novelty of the present findings compared with previous reports (the 2nd major point in my original comments). It is obvious that, by analyzing different sets of SNPs, the authors will find different sets of significant SNP-to-protein associations, but the loci may still be the same – hence, professionally conducted GWAS studies typically report novelty on a locus level and not on a variant level. As the sample is the same, and the proteome data is the same, the authors may find some novelty in the genomic regions that are not covered by the genotyping chip used in the 2018 Science paper. The claim that the comparison of the loci is complicated as they analyze rare and low-frequency variants is totally irrelevant and gives me the impression that the authors may lack expertise in genetic studies. Only at the very end of their response, the authors describe that “restricting to SNPs that are more than 500kb physically distant from the reported lead SNP for the same protein, 1670 SNP-to-protein associations are novel”; this is close to what I was looking for; however, instead of reporting novel SNP-to-protein associations, it would be more relevant to get an idea of the

number of novel loci, and it is unclear if this comparison also included their previous Science paper or just the papers by Sun et al. [1] and Suhre et al. [2]. Also, 500kb distance is rather short (sometimes larger distances, such as 1Mb, are used [3]).

To quickly compare the protein-associated loci in the three studies by the authors, I extracted data from the supplemental files of the 2018 Science paper and the two manuscripts submitted to Nature Communications and produced the attached Manhattan-type plots where each chromosome is plotted separately to allow more clear separation of the key loci; in a typical Manhattan plot, the data points were too densely packed. For each variant, only the most significant association is plotted in order to avoid overlap from the variants showing association with multiple protein. Also, for equality, only associations with p-values below the significance threshold used in the GWAS manuscript ($p < 1.046 \times 10^{-11}$) are plotted instead using the varying study-specific significance thresholds. The plots demonstrate that, even if only a small proportion of the variants was analyzed in all the studies, the association peaks locate on the same genomic regions. To define 'true novel signals' in their study, the authors should identify the protein-associated loci in the present study, in the recently submitted GWAS, in the previous Science paper, and in other previous proteome-GWASs, and carefully address if the loci they find are overlapping or nonoverlapping with the previously reported loci: reproduction of high-impact papers with essentially same findings should not be allowed. It seems that most of the novelty in their manuscripts is in the downstream analyses (i.e., enrichment, colocalization, Mendelian randomization) and not in the genetic associations – the manuscripts should be clearly structured to avoid this false impression (and most likely only one manuscript would be sufficient to report the results of downstream analyses).

Response: It is correct that GWAS studies of "complex disease" report findings at the locus level including the lead SNP and adjacent gene(s), which is not surprising given that lead SNPs rarely point directly to the causal candidate within the genomic region supporting the disease-association of interest. The genetics of gene expression (mRNA, or protein) and its relationship to disease, on the other hand, point directly to the affected target, that is the mRNA or protein, allowing us to better comprehend the pathobiology of complex diseases. Therefore, we respectfully disagree with the reviewer's assertion that "*it would be more relevant to get an idea of the number of novel loci rather than reporting novel SNP-to-protein associations.*" Both are equally important and are not mutually exclusive and will be covered here (see details below). A simple example of why this is important is when two independent SNPs affect the same protein but have opposing effects on the protein, revealing different links to a disease. Information like this can be critical in understanding the pathobiology of complex diseases.

We agree with the reviewer that novelty at the locus level should be considered as well (see below). In general, when estimating novelty, the following factors are considered: 1. A novel independent SNP (pQTL) affecting a novel protein; 2. A novel independent pQTL affecting a protein previously associated with different pQTL(s); 3. A previously known pQTL affecting a novel protein. We now include a detailed comparison of the current study's findings to those of our own GWAS paper by Gudjonsson et al.¹ (NCOMMS-21-24739-T), as well as all proteogenomic studies found in the public domain, including Emilsson et al.² and Sun et al.³ as well as 17 other studies (listed in Supplementary Table S6). For all independent pQTLs, the novelty is now reported at both the SNP-protein and locus-protein levels. Finally, assuming back-to-back publication, we compared the novelty of the **combined results** from the exome-array paper and our GWAS paper by Gudjonsson et al.¹ to all pQTL studies found in the public

domain to date. The results of these analyses are now presented in a new Supplementary Fig. 6, new Supplementary Table 7 and detailed in a new Supplementary Note.

Novel SNP-protein associations: The current study significantly increases the number of genetic signals underlying serum proteins when compared to all 19 external studies found in the public domain (listed in Supplementary Table 6). More specifically, using conditionally independent study-wide significant associations (Supplementary Table 2) and an LD threshold of $r^2 < 0.5$ for novel associations, the current study reveals 76.8% novel SNP-protein associations compared to Emilsson et al.², 75.5% novel SNP-protein associations compared to Sun et al.³, and 59.3% novel SNP-protein associations compared to all published pQTL studies (see new Supplementary Fig. 6a). The LD threshold of $r^2 < 0.9$ is also shown for this comparison (Supplementary Note and Supplementary Fig. 6a, right panel). Similarly, when we compared all conditionally independent study-wide significant SNP-protein associations in the new GWAS paper by Gudjonsson et al.¹ with the current exome-array study, using LD of $r^2 < 0.5$ for novel associations, we find that 49.5% (2053 of 4147 independent SNP-protein associations in the GWAS) were GWAS-specific, while 48.4% (1937 of 4001 independent SNP-protein associations in the exome-array study) were exome-array-specific (left panel in Supplementary Fig. 6b). Using LD of $r^2 < 0.9$ for the comparison, 59.8% were GWAS specific while 58.6% were exome-array specific (right panel in Supplementary Fig. 6b). These comparisons imply that the two studies, as separate investigations, complement each other well. Finally, we obtain 6362 SNP-to-protein associations by combining all unique and common SNP-to-protein signatures from both companion studies. When compared to external data sets using LD of $r^2 < 0.5$ and $r^2 < 0.9$ for novel associations, we find that 60.0% and 64.8%, respectively, of the conditionally independent SNP-protein associations presented by our two companion papers are novel (see new Supplementary Fig. 6c). More details of these comparisons are provided in the Supplementary Note.

a

b

c

Supplementary Fig. 6. a. The comparison of all conditionally independent SNP-to-protein associations in the exome-array paper to all 19 external studies found in the public domain (listed in Supplementary Table 6) including for instance Sun et al. (2018)³ and Emilsson et al (2018)². The label -Other- refers to any proteogenomic study in the

public domain that is not Sun et al. (2018)³ or Emilsson et al (2018)². The term -Any- refers to all 19 proteogenomic studies that have been published to date (Supplementary Table 6). Two different LD thresholds were used for this comparison: LD of $r^2 < 0.5$ (left panel) and $r^2 < 0.9$ (right panel). **b.** All conditionally independent study-wide significant SNP-to-protein associations in the current exome-array were compared to those reported in our companion GWAS paper¹ at two different LD thresholds: LD of $r^2 < 0.5$ (left panel) and $r^2 < 0.9$ (right panel). **c.** Each companion (GWAS and exome-array) study's combined unique and common independent study-wide significant pQTLs were compared to published proteogenomic studies for novelty at different LD thresholds: $r^2 < 0.5$ (left panel) and $r^2 < 0.9$ (right panel). The label -Other- refers to any proteogenomic study in the public domain (Supplementary Table 6) that is not Sun et al. (2018)³ or Emilsson et al (2018)². The term -Any- refers to all 19 proteogenomic studies that have been published to date (Supplementary Table 6). The barplots in **a.-c.** indicates whether or not a matching pQTL association has previously been reported (known) or not (novel).

Novel locus-protein associations: First, we examined which aptamers had a study-wide significant signal in each of the two companion papers, determining how many are explicitly found in each paper and how many are found in both. We then defined neighboring lead SNP signals to be from the same locus if the distance between the signals was less than 300kb, which is consistent with the window used in our GWAS paper¹ and our previous publication in Science². More specifically, for each study we combined neighboring independent lead SNP signals into a unified locus until no other SNP signal was within 300kb of the locus, at which point we define a new locus and proceed in the same way. The output of this procedure were two sets of genomic ranges, one for each paper, which we then analyzed to see how many loci overlapped and how many were unique between the two studies. When comparing the papers to previously published proteomic studies, we combined the previous studies into one dataset, performed the same operation on this larger dataset and compared genomic ranges thus obtained to the previously mentioned ranges.

When a locus in one study overlaps with a locus in the other, we consider it shared (independent of if they associate with the same proteins or not). This analysis reveals that the GWAS study by Gudjonsson et al.¹ does not cover 321 of the 881 loci identified in the present exome-array paper. When the exome array data is compared to the results of the previous 19 pQTL studies (listed in Supplementary Table 6), the current study finds 292 novel loci. Finally, when the current exome array study and the study by Gudjonsson et al.¹ are combined and compared to all previously published pQTL studies, the two studies yielded 404 novel loci. These findings have been summarized in a new Supplementary Table 7. Next, we looked at locus-protein associations and considered them shared if the locus overlapped with a locus in the other study that was furthermore associated with the same protein. In this study, 762 of 3103 locus-protein associations are unique to the exome-array study when compared to the GWAS paper by Gudjonsson et al.¹. When the exome array results are compared to all previous pQTL publications (listed Supplementary Table 6), 1473 locus-protein associations were found to be novel. Similarly, when compared to previous pQTL publications (Supplementary Table 6), the current study and the GWAS by Gudjonsson et al.¹ combined revealed 1950 novel locus-protein associations. These findings have been summarized in a new Supplementary Table 7 (see below) and detailed in a new Supplementary Note. In conclusion, the exome array yields many novel findings at both the locus-protein and SNP-protein levels.

Study	Compared to	Loci/regions per study in column A			Locus - protein interactions per study in column A		
		Total	Shared	Novel	Total	Shared	Novel
Current exome array study	Companion GWAS (Gudjonsson et al.)	881	560	321	3103	2341	762
Current exome array study	All pQTLs in the public domain (Supplementary Table 6)	881	589	292	3103	1630	1473
Companion GWAS (Gudjonsson et al.) + exome array study	All pQTLs in the public domain (Supplementary Table 6)	1097	693	404	3845	1895	1950

Supplementary Material now includes a Supplementary Note titled "Estimates of novelty for pQTLs reported in the current study" that includes the following description:

"Novel SNP-protein associations: In general, when estimating novelty of SNP-protein associations, the following factors are considered: 1. A novel independent SNP (pQTL) affecting a novel protein; 2. A novel independent pQTL affecting a protein previously associated with different pQTL(s); 3. A previously known pQTL affecting a novel protein. In comparison to all 19 external studies found in the public domain (listed in Supplementary Table 6), the current study significantly increases the number of genetic signals underlying serum proteins. More to the point, using conditionally independent study-wide significant associations (Supplementary Table 2) and LD threshold of $r^2 < 0.5$ for novel associations, the current study reveals 76.8% novel SNP-protein associations compared to Emilsson et al.², 75.5% compared to Sun et al.³, and with 59.3% of the 4001 SNP-protein associations being novel in comparison to all published pQTL studies (left panel in Supplementary Figure 6a). These comparisons are also shown using LD threshold of $r^2 < 0.9$ for novel associations. Here, the present study finds 81.7% novel SNP-protein associations compared to Emilsson et al.², 76.2% compared to Sun et al.³, and 62.0% novel SNP-protein associations compared to all published pQTL studies (Supplementary Fig. 6a, right panel).

Similarly, when we compared all conditionally independent study-wide significant SNP-to-protein associations in the new GWAS paper by Gudjonsson et al.¹ with the current exome-array study, using LD of $r^2 < 0.5$ for novel associations, we find that 49.5% (2053 of 4147 conditionally independent SNP-protein associations in the GWAS) were GWAS-specific, while 48.4% (1937 of 4001 conditionally independent SNP-protein associations in the exome-array study) were exome-array-specific (left panel in Supplementary Fig. 6b). Using LD of $r^2 < 0.9$ for the comparison, 59.8% were GWAS specific while 58.6% were exome-array specific (right panel in Supplementary Fig. 6b).

Finally, we obtain 6362 SNP-protein associations by combining all unique and common SNP-protein signatures from both companion studies. Here, at LD of $r^2 < 0.5$ for novel SNP-protein associations, 77.9% were novel compared to Emilsson et al.² and 74.6% compared to Sun et al.³, while at LD of $r^2 < 0.9$, 82.3% were novel compared to Emilsson et al.² and 77.4% compared to Sun et al.³. When compared to external data sets with LD of $r^2 < 0.5$ and $r^2 < 0.9$, we find that 60.0% and 64.8%, respectively, of the conditionally independent SNP-protein associations presented by our two companion papers are novel (Supplementary Fig. 6c).

Novel locus-protein associations: First, we examined which aptamers were study-wide significant in each of the two companion papers, determining how many are explicitly found in each paper and how many are found in both. We then defined neighboring lead SNP signals to be from the same locus if the distance between the signals was less than 300kb, which is consistent with the window used in our GWAS paper¹ and our previous publication in Science². More specifically, for each study we combined neighboring conditionally independent lead SNP signals into a unified locus until no other SNP signal was within 300kb of the locus, at which point we define a new locus and proceed in the same way. The output of this procedure were two sets of genomic ranges, one for each paper, which we then analyzed to see how many loci overlapped and how many were unique between the two studies. When comparing the papers to previously published proteomic studies, we combined the previous studies into one dataset,

performed the same operation on this larger dataset and compared genomic ranges thus obtained to the previously mentioned ranges.

When a locus in one study overlaps with a locus in the other, we consider it shared (independent of if they associate with the same proteins or not). This analysis reveals that the GWAS study by Gudjonsson et al.¹ does not cover 321 of the 881 loci identified in the present exome-array paper (Supplementary Table 7). When the exome array data is compared to the results of the previous 19 pQTL studies (listed in Supplementary Table 6), the current study finds 292 novel loci (Supplementary Table 7). Finally, when the current exome array study and the study by Gudjonsson et al.¹ are combined and compared to all previously published pQTL studies, the two studies yield 404 novel loci (Supplementary Table 7). Next, we looked at locus-protein associations and considered them shared if the locus overlapped with a locus in the other study that was furthermore associated with the same protein. In this study, 762 of 3103 locus-protein associations are unique to the exome-array study when compared to the GWAS paper by Gudjonsson et al.¹. When the exome array results are compared to all previous pQTL publications (Supplementary Table 6), 1473 locus-protein associations were found to be novel (Supplementary Table 7). Similarly, when compared to previous pQTL publications, the present study and the study by Gudjonsson et al.¹ combined revealed 1950 novel locus-protein associations (Supplementary Table 7). In conclusion, the exome array yields many novel findings at both the locus-protein and SNP-to-protein levels.”

On page 9, sentences 209–231, we have highlighted the exome-array study's novel findings in comparison to our own GWAS companion paper and all published pQTL studies todate:

” We outlined the construction of the serum protein network in our previous report and identified common genetic variants underlying the network structure⁴. This included a targeted study of the effects of common cis and cis-to-trans acting variants on levels of serum proteins. Previously, we discovered that 80% of cis pQTL effects and 74% of trans pQTL effects were replicated across populations and proteomics platforms measuring common variants⁴. We estimated the novelty of pQTL findings reported in the present study at both SNP-protein and locus-protein levels (see Supplementary Note for details). In brief, using all conditionally independent study-wide significant associations (Supplementary Table 2) and a linkage disequilibrium (LD) threshold of $r^2 < 0.5$ for novel associations, the current study's SNP-protein associations are 76.8% novel compared to Emilsson et al.⁴, 75.5% novel compared to Sun et al.¹¹, and 59.3% novel compared to all published pQTL studies (Supplementary Fig. 6a and Supplementary Note). Similarly, in comparison to our companion GWAS paper¹ and using the same LD threshold for novel associations, we find that 48.4% were exome-array-specific (Supplementary Fig. 6b and Supplementary Note). By combining all unique and common SNP-protein signatures from both companion studies, we obtain 6362 SNP-protein associations, of which 60.0% (at LD threshold of $r^2 < 0.5$) are novel when compared to external pQTL datasets (Supplementary Note and Supplementary Fig. 6c). Finally, when estimating novelty at the locus-protein level, we find that 321 loci and 762 locus-protein associations in the current study are novel compared to our companion paper¹ (Supplementary Table 7 and Supplementary Note). When the two companion studies were combined, they yielded 404 new loci and 1950 new locus-protein associations, which were not found in previous pQTL publications (Supplementary Table 7 and Supplementary Note).”

As regards the size of a *cis*-acting window in genetics of gene/protein expression studies, in the literature it is seemingly chosen arbitrarily. In other words, the size of the window varies significantly depending on the research group identifying *cis* acting effects. The main reason we did not use a 1Mb region in this instance is simple: the exome array platform is enriched for structural variants located in the protein encoding gene of interest, so as close to the target as possible. As previously mentioned, we have used a 300kb *cis*-acting window for the exome-array paper, which is consistent with the window used in our GWAS paper (NCOMMS-21-24739-T) and our previous publication in Science².

In addition, the following issues gave me the impression that the authors have not finalized the manuscript in a manner that would be appropriate to a high-impact journal:

– In my original Minor Comment 5, I asked to include LD information in the regional association plots. Public data is not ideal for this purpose, especially for rare/low-frequency variants, which can be seen as missing LD information in all the regional association plots. Preferably, the authors should calculate LD based on their genetic data and plot the regional associations using software that allows implementation of your own LD information. Also, the inclusion of rs-IDs of the key variants in the regional association plots would aid the reader to follow the discussion of the findings; for example, now the reader can only assume that rs75932628 and rs610932 are the lead variants in the TREM2-associated loci, as this is not shown in the figures and not clearly indicated in the text.

Response: We thank the reviewer for his/her suggestion and have replaced the regional association plots with new ones based on our LD-based genetic data and highlighted the key SNPs of interest. The new LocusZoom plots are shown below.

a

b

Supplementary Fig. 3. TREM2 regional plots (LocusZoom) based on exome array variants at **a.** chromosome 6 and **b.** chromosome 11, using LD data from the AGES-RS cohort. Each plot highlights study-wide significant pQTLs.

a**b**
Supplementary Fig. 5. SVEP1 regional plots (LocusZoom) based on exome array variants at **a.** chromosome 1 and **b.** chromosome 9, using LD data from the AGES-RS cohort. Each plot highlights study-wide significant pQTLs.

– It is impossible to read VEP results in Supplementary Table 3. The authors should see more efforts to ease the reader

Response: For clarification, we have revised the VEP results table format in Supplementary Table 3. Also, see our response to the revision of column headers for all Supplementary Tables.

– I am not sure why rs704 (chr17) that associates ($p \sim 1e-12$) with TREM2 level (Supplementary Tables 1 and 2) is not plotted in Figure 3a and is not mentioned in the text with the TREM2 associations in chromosomes 6 and 11.

Response: We appreciate the reviewer pointing this out, but the confusion stems from the fact that there are two aptamers that target TREM2, one of which (aptamer 5635-66) is significantly associated with the LOAD variant at chr. 6, as shown in Supplementary Tables 1 and 2. Both aptamers were associated with the LOAD related region on chr. 11. The aptamer that was associated to both LOAD-related loci was highlighted in the main text.

To clarify, we have added the following information to Fig. 3a's legend:

"The Manhattan plot highlights variants at two distinct chromosomes associated with serum TREM2 (aptamer 5635-66, Supplementary Table 2) levels."

– Figure legends should describe the methodology used for producing the figure so that the figures are self-explanatory. In this manuscript, figure legends tend to describe results. Key information is missing, including the y-axis units in Figures 3b, 4b, 4c, 5a, and 5d. Are the boxplots shown for adjusted protein levels?

Response: We have now improved the description of all units, as well as the method of normalization and adjustments, in all Figure legends.

Figure 3b, for example, now includes the following text:

"The x-axis of each box plot shows the genotypes for the corresponding protein-associated SNP, while the y-axis denotes the Box Cox transformed, age and sex adjusted serum protein levels"

And for Figure 4c:

"The x-axis of the box plots shows the health status of individuals, while the y-axis denotes the Box Cox transformed, age and sex adjusted serum protein levels"

– Supplementary Figure 4: The legend says "The correlation matrix's upper triangle depicts the beta-values", but the color map refers to 'Spearman correlation' for which the coefficient is called ρ (rho), not beta.

Response: The reviewer is entirely correct, and we have updated the legend to Supplementary Figure 4 to reflect this, as well as taken advantage of the opportunity to add this detail to legends for Supplementary Figure 2.

– The authors should go through the content in the Supplemental Tables and check if all column headers are explained sufficiently (for example, I would assume that columns ‘Gene’ and ‘EntrezGeneSymbol’ may cause confusion) and if all the columns need to be repeated in all the tables or if overlapping information could be reduced.

Response: We have now re-annotated each column header for clarity and consistency across the Supplementary Tables. This included, for example, avoiding abbreviations and using “Protein target” instead of “Gene or Gene symbol”. The gene containing the exome array variant is denoted by the column title Gene. This comment is also related to the reviewer’s comment about the annotation of the VEP results (see above), as well as the reference to pQTL predicted functional consequences (see comment below on missense mutations in SVEP1). We have updated the legends on all supplementary tables for clarity.

– Lines 159-161: “Overall, we found eight different missense mutations in SVEP1 that were associated with SVEP1 serum levels (Supplementary Table 1).” Gene name and mutation type are not given in Supplementary Table 1. Is there information missing from this table, or is the reference to the wrong table?

Response: We thank the reviewer for bringing this to our attention and apologize for not revising the sentence for the previous updated revision. Due to the fact that this was written before we performed the conditional analysis, we now refer to Supplementary Table 2 for independent pQTLs regulating SVEP1. Also, Supplementary Table 2 now includes a new column highlighting the consequence of any SNP, and this sentence now reads correctly as follows:

On page 7, lines 165-167:

“In total, we found four conditionally independent missense mutations in SVEP1 that were linked to serum SVEP1 levels (Supplementary Table 2).”

In general, all counts in the text have been carefully revised with references to the supplementary tables, and we have highlighted unique protein Entrez annotations as well as the number of aptamers underlying protein counts. However, we should note that the Bonferroni adjusted P-value threshold was set based on the number of aptamers (4782) rather than the number of unique human proteins (4137).

– Reporting of Mendelian randomization results fall somewhat short of the state-of-art. A thorough report would provide the results also in a tabulated form (in the supplement), p-values for horizontal pleiotropy and heterogeneity, and also plots for leave-one-out sensitivity analyses. In the description of the methods, the authors should describe the reference population for calculating LD. Further, by running bidirectional Mendelian randomization, the authors would have an opportunity to distinguish disease biomarkers from causative agents.

Response: We agree with the reviewer and have therefore re-examined the entire MR analysis. For this study, we used the AGES-RS as the reference population to compute the LD structure. Because we are using our own population-based LD structure, the number of instruments used may differ from previous results in some cases. We now include MR-Egger regression to assess

horizontal pleiotropy and Cochran's Q-statistics to estimate heterogeneity. A new Supplementary Table 5 has been created to numerically report all positive and negative results. Below are the new scatter plots for TREM2, SVEP1, and ASIP. Again, we prefer to use only *cis*-acting genetic instruments, but in the case of TREM2, which has a single *cis*-acting pQTL, we also used *trans*-acting pQTL genetic instruments. The MR-Egger sensitivity analysis revealed no evidence of horizontal pleiotropy for the different causal tests (Supplementary Table 5). Given the physical distance (different chromosomes) between the *cis* and *trans* acting TREM2 instruments, the Cochran's Q-value was significant (Supplementary Table 5), indicating heterogeneity, as expected. A bi-directional MR analysis, as suggested by the reviewer, was attempted but abandoned due to a lack of overlapping SNPs between the exome array and the GWAS for the various outcomes.

Finally, we now include Leave-One-Out plots as a new Supplementary Figure 7 for all the established causal relationships, revealing that the causal estimate was not reliant on any single genetic instrument (see below).

We have included a new text in the Method section:

*“We applied the “TwoSampleMR” R package⁵ to perform a two-sample MR analysis to test for causal associations between protein and outcome (protein-to-outcome). For different outcomes we used GWAS associations for LOAD in Europeans⁶, malignant melanoma in European individuals from the UK biobank data (UKB-b-12915)⁷, T2D in Europeans⁸, CHD in Europeans⁹ and systolic blood pressure in Europeans¹⁰. Genetic variants (SNPs) associated with serum protein levels at a genome-wide significant threshold ($P < 5 \times 10^{-8}$) identified in the AGES dataset and filtered to only include uncorrelated variants ($r^2 < 0.2$) were used as instruments. More to the point, genetic instruments within the *cis* window (150kb up- and downstream of and including the encoding gene) for each aptamer were then clumped such that variants in high LD ($r^2 \geq 0.2$) were combined, using the LD structure in the AGES population. The inverse variance weighted (IVW) method¹¹ was used for the MR analysis, with P-values < 0.05 considered significant. For sensitivity analyses we used the intercept term from MR Egger regression¹² to determine whether there was evidence of horizontal pleiotropy, and Cochran's Q-statistic¹³ to evaluate heterogeneity of genetic instruments. A leave-one-out analysis was also carried out to assess if the observed causal estimates relied on any single SNP instrument. A bi-directional MR analysis was also attempted but abandoned due to the lack of overlapping SNPs between the exome array and the GWAS for the different outcomes.”*

We have added the following text to page 10, lines 253-255:

“These analyses found no evidence of horizontal pleiotropy (Supplementary Table 5), nor did they demonstrate that the causal estimates were dependent on a single genetic instrument (Supplementary Fig. 7)”

Fig. 3d. Scatterplot for the TREM2 protein supported as having a causal effect on LOAD in a two-sample MR analysis. The figure demonstrates the estimated effects (with 95% confidence intervals) of their respective *cis*- and *trans*-acting genetic instruments on the serum TREM2 levels in AGES-RS (x-axis) and risk of LOAD through a GWAS by Kunkle et al.⁶ (y-axis), using 21,982 LOAD cases and 41,944 controls. The broken line indicates the inverse variance weighted causal estimate ($\beta = -0.240$, $SE = 0.059$, $P = 5.3 \times 10^{-5}$), while the dotted line shows the MR-Egger regression (see Supplementary Table 5).

Fig. 4e. Scatterplot for the SVEP1 protein supported as having a causal effect on T2D in a two-sample MR analysis. The figure demonstrates the estimated effects (with 95% confidence intervals) of the SNP effect on serum SVEP1 levels and T2D from a GWAS in Europeans⁸ (y-axis), with 74,124 T2D patients and 824,006 controls. The broken line indicates the inverse variance weighted causal estimate ($\beta = 0.104$, $SE = 0.023$, $P = 5.7 \times 10^{-6}$), while the dotted line demonstrates the MR-Egger regression (see Supplementary Table 5).

Fig. 5b. Scatterplot for the ASIP protein supported as having a causal effect on malignant melanoma in a two-sample MR analysis. The figure demonstrates the estimated effects (with 95% confidence intervals) of their respective genetic instruments on the serum ASIP levels in AGES (x-axis) and risk of melanoma in GWAS by UK biobank data (UKB-b-12915)⁷ (y-axis), that included 3598 melanoma cases and 459,335 controls. The broken line indicates the inverse variance weighted causal estimate ($\beta = 0.0024$, $SE = 0.0003$, $P = 1.1 \times 10^{-17}$), while the dotted line shows the MR-Egger regression (see Supplementary Table 5).

a

b

c

Supplementary Fig. 7. Leave-one-out plots for the MR analyses of **a.** TREM2 (one *cis* plus four *trans* pQTLs), **b.** SVEP1 (six *cis* pQTLs) and **c.** ASIP (four *cis* pQTLs) to assess if the causal estimates are reliant on any single SNP instrument for a given MR test. The y-axis denotes the individual pQTL instruments, while the x-axis denotes the effect size (beta-values).

– Lines 108-110: “This suggests that proteins bound for the systemic environment are subject to more genetic regulation than other proteins identified by the current platform.” – Do you mean to say that the circulating level/quantity of non-secreted proteins is under stricter genetic control than the quantity of secreted proteins? If so, it is problematic that ‘quantity’ is not mentioned, as surely all endogenous proteins are under genetic regulation. Also, it is confusing that the terminology is different from the preceding sentence (does “proteins bound for the systemic environment” refer to non-secreted proteins?).

Response: We apologize for any confusion caused by this sentence, which has been revised for clarity. In fact, we're referring to proteins that are secreted. The sentence is now as follows:

On page 5, lines 110-111:

“This implies that secreted proteins are subject to different, and possibly stronger, genetic control than other proteins identified by the current platform.”

– Figure 4: Why is the scatter plot presented only for T2D but not CHD and systolic BP? Those could be in the supplement if not in the manuscript main body.

Response: We refer back to our previous response to a similar MR analysis comment. Because the links to systolic blood pressure and CHD were not significant for SVEP1, we only highlighted the causal link to T2D for SVEP1. However, all MR analysis outcomes, both positive and negative, are now available in a new Supplementary Table 5, including the lack of a causal relationship between SVEP1 and CHD and systolic blood pressure. To emphasize the negative results, a reference to Supplementary Table 5 has been added to the main text.

– Lines 130-132 and lines 157-159; I am not sure if I follow the justification for the selection of the key variants presented in the examples. For example, in case of TREM2 association in chr11, the authors highlight an intergenic variant rs610932 (pTREM2.5635-66=1e-106; pTREM2.11851-21=3e-46) instead of rs7232, a missense variant with deleterious prediction (pTREM2.5635-66=8e-131; pTREM2.11851-21=2e-54) that is also included as an instrument in Mendelian randomization. I understand the prior link with LOAD, but isn't this exactly where the authors would have an opportunity to highlight the advantage of their exonic data? Perhaps they could even suggest something along the lines that “the previously reported association with LOAD in this locus could arise due to coding variants in MS4A6A, such as rs7232, a missense variant with deleterious prediction in LD ($r^2 \sim 0.7$) with the LOAD lead variant rs610932” as, based on Supplementary Table S2, the association of rs610932 does not remain significant in the conditional analyses?

Response: We thank the reviewer for making a good point here. In fact the MS4A cluster has recently been shown to modulate production of soluble TREM2¹⁴. As regards rs7232, the variant, just like the 3' UTR variant rs610932, has previously been associated with LOAD (see Supplementary Table 4). To emphasize this point we have added the following sentence:

On page 6, lines 152-156:

”The genetic instrument rs7232 (Fig. 3d), an independent variant associated with TREM2 (Supplementary Table 2), is a missense variant in MS4A6A that has previously been linked to LOAD (Supplemental Table 4), but the MS4A cluster has recently been shown to modulate production of soluble TREM2¹⁴. This could imply that the variant is directly involved in the pathogenesis of LOAD.”

References

- 1 Gudjonsson, A. *et al.* A genome-wide association study of serum proteins reveals shared loci with common diseases. *bioRxiv*, 2021.2007.2002.450858, doi:10.1101/2021.07.02.450858 (2021).
- 2 Emilsson, V. *et al.* Co-regulatory networks of human serum proteins link genetics to disease. *Science* **361**, 769-773, doi:10.1126/science.aag1327 (2018).
- 3 Sun, B. B. *et al.* Genomic atlas of the human plasma proteome. *Nature* **558**, 73-79, doi:10.1038/s41586-018-0175-2 (2018).
- 4 Emilsson, V. *et al.* Co-regulatory networks of human serum proteins link genetics to disease. *Science* **361**, 769-773, doi:10.1126/science.aag1327 (2018).
- 5 Hemani, G. *et al.* The MR-Base platform supports systematic causal inference across the human phenome. *Elife* **7**, doi:10.7554/eLife.34408 (2018).
- 6 Kunkle, B. W. *et al.* Genetic meta-analysis of diagnosed Alzheimer's disease identifies new risk loci and implicates A β , tau, immunity and lipid processing. *Nat Genet* **51**, 414-430, doi:10.1038/s41588-019-0358-2 (2019).
- 7 Sudlow, C. *et al.* UK biobank: an open access resource for identifying the causes of a wide range of complex diseases of middle and old age. *PLoS Med* **12**, e1001779, doi:10.1371/journal.pmed.1001779 (2015).
- 8 Mahajan, A. *et al.* Fine-mapping type 2 diabetes loci to single-variant resolution using high-density imputation and islet-specific epigenome maps. *Nat Genet* **50**, 1505-1513, doi:10.1038/s41588-018-0241-6 (2018).
- 9 Nikpay, M. *et al.* A comprehensive 1,000 Genomes-based genome-wide association meta-analysis of coronary artery disease. *Nat Genet* **47**, 1121-1130, doi:10.1038/ng.3396 (2015).
- 10 Evangelou, E. *et al.* Genetic analysis of over 1 million people identifies 535 new loci associated with blood pressure traits. *Nat Genet* **50**, 1412-1425, doi:10.1038/s41588-018-0205-x (2018).
- 11 Burgess, S., Butterworth, A. & Thompson, S. G. Mendelian randomization analysis with multiple genetic variants using summarized data. *Genet Epidemiol* **37**, 658-665, doi:10.1002/gepi.21758 (2013).
- 12 Burgess, S. & Thompson, S. G. Interpreting findings from Mendelian randomization using the MR-Egger method. *Eur J Epidemiol* **32**, 377-389, doi:10.1007/s10654-017-0255-x (2017).
- 13 Bowden, J. *et al.* Improving the accuracy of two-sample summary-data Mendelian randomization: moving beyond the NOME assumption. *Int J Epidemiol* **48**, 728-742, doi:10.1093/ije/dyy258 (2019).
- 14 Deming, Y. *et al.* The MS4A gene cluster is a key modulator of soluble TREM2 and Alzheimer's disease risk. *Sci Transl Med* **11**, doi:10.1126/scitranslmed.aau2291 (2019).

Reviewers' Comments:

Reviewer #3:

Remarks to the Author:

To indicate novelty in their study, the authors have identified 881 loci using a 300 kb distance that they describe in the response and in the Supp. Table 7 – however, the authors have not added the information to the manuscript text. I also hoped that the authors would clearly indicate the novel loci, as done in the proteome study by Sun et al.[1] (Supplementary Table 4, column 'Previously reported') rather than providing only a summary table.

I also remain in my original statement saying that the 300 kb distance is rather short, and I would have preferred the use of 1Mb distance as is done, for example, in the proteome study by Sun et al.[1] and exome-wide study of plasma lipids by Liu et al.[2]. The authors argue that “The main reason we did not use a 1Mb region in this instance is simple: the exome array platform is enriched for structural variants located in the protein encoding gene of interest, so as close to the target as possible”. The reason to use larger distances is to ensure that independent loci are reported regardless of whether exome or genome-wide data were analyzed – in fact, in some occurrences, the genomic loci are identified depending on LD structure[3] rather than physical distance. But as there are varying practices, I could probably let it slide.

Also, I would like to give the following comment regarding the importance of reporting novelty on locus rather than on variant level: Without further functional validation of the variants in the associated regions, it is impossible to identify the true causal variants regardless of the study phenotype (i.e., a “complex disease” or a gene expression product) or the genotype data used. Due to the genome structure, you will identify multiple associated variants that are in LD with the variants causing the variation in the phenotype, and the more variants you test, the more associations you will find. This is well demonstrated in your conditional analyses: for TREM2 association in chr11 near MS4A6A you report 10 significant SNPs (Supp. Table 1), and only 2 of those are independent (Supp. Table 2). I would like to note that, obviously, it will be necessary to report SNP-to-protein associations as the authors have done, but I find that in terms of reporting novelty, locus-level findings are more valuable, as those provide more information on the genetic pathways involved in the regulation of circulating protein levels.

It remains for the Editors to decide whether the novelty is enough for three high-impact publications, but I question it for the reasons already mentioned (a single population, no replication; the overlap between studies remains unclear – of note, it clearly says in the Supplemental Methods of the Science paper[4] that “genotypes assayed through the exome-wide genotyping array Illumina HumanExome Beadchip were available for all the 5,457 AGES subjects” even if in their original response letter the authors give an impression that Illumina 370CNV was used in the previous study). I have no further comments.

References

1. Sun, B. B. et al. Genomic atlas of the human plasma proteome. *Nature* 558, 73–79 (2018).
2. Liu, D. J. et al. Exome-wide association study of plasma lipids in >300,000 individuals. *Nat. Genet.* 49, 1758–1766 (2017).
3. Watanabe, K., Taskesen, E., Van Bochoven, A. & Posthuma, D. Functional mapping and annotation of genetic associations with FUMA. *Nat. Commun.* 8, 1–11 (2017).
4. Emilsson, V. et al. Co-regulatory networks of human serum proteins link genetics to disease. *Science* (80-.). 361, 1–61 (2018).

We are pleased to submit our revised manuscript now entitled “*Coding and regulatory variants associated with serum protein levels and disease*” (NCOMMS-20-16689C). We have responded to reviewer 3's additional comments/suggestions below.

Responses below are provided in blue font. Text added to the revised manuscript has been italicized. Page and page numbers listed below refer to the position of the new or modified text in the *clean* version of the revised manuscript.

REVIEWERS' COMMENTS

Reviewer #3 (Remarks to the Author):

To indicate novelty in their study, the authors have identified 881 loci using a 300 kb distance that they describe in the response and in the Supp. Table 7 – however, the authors have not added the information to the manuscript text. I also hoped that the authors would clearly indicate the novel loci, as done in the proteome study by Sun et al.¹ (Supplementary Table 4, column ‘Previously reported’) rather than providing only a summary table.

Response: We thank the reviewer for bringing this to our attention, and we have clarified the information detailed in the Supplementary Note and Supplementary Data 7 regarding the locus-level novelty estimate in the main text (Page 9, lines 228 to 230). In terms of presenting the SNP-to-protein and locus-protein novelty, we used the same expression as in our companion GWAS paper², namely as a new Supplementary Note, Supplementary Figure 6, and Supplementary Data 7.

“Finally, when estimating novelty at the locus-protein level, we find that 321 out of 881 loci and 762 out of 3103 locus-protein associations identified in the current study are novel compared to our companion paper² (Supplementary Data 7 and Supplementary Note)”

I also remain in my original statement saying that the 300 kb distance is rather short, and I would have preferred the use of 1Mb distance as is done, for example, in the proteome study by Sun et al.¹ and exome-wide study of plasma lipids by Liu et al.³. The authors argue that “*The main reason we did not use a 1Mb region in this instance is simple: the exome array platform is enriched for structural variants located in the protein encoding gene of interest, so as close to the target as possible*”. The reason to use larger distances is to ensure that independent loci are reported regardless of whether exome or genome-wide data were analyzed – in fact, in some occurrences, the genomic loci are identified depending on LD structure⁴ rather than physical distance. But as there are varying practices, I could probably let it slide.

Response: We appreciate the reviewer's thoughts and opinions on this matter, but we believe that a narrower window is more applicable to the exome-focused array with sparse genotype information.

Also, I would like to give the following comment regarding the importance of reporting novelty on locus rather than on variant level: Without further functional validation of the variants in the associated regions, it is impossible to identify the true causal variants regardless of the study phenotype (i.e., a “complex disease” or a gene expression product) or the genotype data used. Due to the genome structure, you will identify multiple associated variants that are in LD with the variants causing the variation in the phenotype, and the more variants you test, the more associations you will find. This is well demonstrated

in your conditional analyses: for TREM2 association in chr11 near *MS4A6A* you report 10 significant SNPs (Supp. Table 1), and only 2 of those are independent (Supp. Table 2). I would like to note that, obviously, it will be necessary to report SNP-to-protein associations as the authors have done, but I find that in terms of reporting novelty, locus-level findings are more valuable, as those provide more information on the genetic pathways involved in the regulation of circulating protein levels.

Response: We appreciate the reviewer's thoughts and opinions on this matter.

It remains for the Editors to decide whether the novelty is enough for three high-impact publications, but I question it for the reasons already mentioned (a single population, no replication; the overlap between studies remains unclear – of note, it clearly says in the Supplemental Methods of the Science paper¹ that “genotypes assayed through the exome-wide genotyping array Illumina HumanExome Beadchip were available for all the 5,457 AGES subjects” even if in their original response letter the authors give an impression that Illumina 370CNV was used in the previous study). I have no further comments.

Response: We apologize for not clarifying this to the reviewer in our previous response. It is correct that we mentioned Illumina HumanExome Beadchip exome array genotypes were available for all 5457 AGES subjects in the Supplementary Information to our Science paper⁵. However, for that study, common genotypes from the Illumina Hu370CNV Array platform, rather than from the exome array platform, were used for a subset of 3200 subjects. In other words, while the exome array platform was mentioned, it was not investigated for that paper. We thank the reviewer for bringing this to our attention and apologize for the ambiguity in our previous Science Supplementary Material. This was more of a statement describing the material and should not have been mentioned in the Supplementary Material, and it may explain a lot of the reviewer's previous comments in hindsight.

References

- 1 Sun, B. B. *et al.* Genomic atlas of the human plasma proteome. *Nature* **558**, 73-79, doi:10.1038/s41586-018-0175-2 (2018).
- 2 Gudjonsson, A. *et al.* A genome-wide association study of serum proteins reveals shared loci with common diseases. *bioRxiv*, 2021.2007.2002.450858, doi:10.1101/2021.07.02.450858 (2021).
- 3 Liu, D. J. *et al.* Exome-wide association study of plasma lipids in >300,000 individuals. *Nature Genetics* **49**, 1758-1766, doi:10.1038/ng.3977 (2017).
- 4 Watanabe, K., Taskesen, E., van Bochoven, A. & Posthuma, D. Functional mapping and annotation of genetic associations with FUMA. *Nat Commun* **8**, 1826, doi:10.1038/s41467-017-01261-5 (2017).
- 5 Emilsson, V. *et al.* Co-regulatory networks of human serum proteins link genetics to disease. *Science* **361**, 769-773, doi:10.1126/science.aag1327 (2018).